# The dorsal aortic compartment is a developmental source of brown adipose tissue in mice

Sophie Heider [1,2], Cornelius Fischer[3,4], Ali Kerim Secener [1,3,4], Pedro Vallecillo-García[1,5], Georgios Kotsaris[1], Zarah G. Meisen[1,6], Verena Pawolski[1], Claudia Giesecke-Thiel [7], Thomas Conrad [3,4], Tim J. Schulz [2], Sascha Sauer [3,4] & Sigmar Stricker [1,6] ✉

White adipose tissue primarily stores energy while brown adipose tissue dissipates energy as heat, holding promise for therapeutic use. Brown adipose tissue in the anterior trunk is believed to derive from the somitic mesoderm, although some depots are of partially unknown origin. Here we show that the subscapular, lateral, cervical and peri-aortic brown adipose depots, but not the interscapular depot, are in part formed by a non-somitic source. Single-cell sequencing along with genetic lineage tracing indicates that at embryonic day 9.5 the dorsal aorta compartment harbors multipotent mesenchymal progenitors expressing the transcription factor Osr1. Spreading laterally from the dorsal aortic midline, these cells contribute to adipose, cartilage and myogenic lineages. This study uncovers an alternative source of brown adipose tissue and suggests that a fraction of dorsal aorta-associated mesenchymal Osr1+ cells may represent the in vivo correlate of a multipotent progenitor cell type so far only characterized in vitro, the mesoangioblast.

In light of the obesity pandemic, metabolic health is a significant issue in modern society. Eutherian mammals harbor two types of adipose tissue (AT), white and brown adipose tissue (WAT and BAT, respectively). WAT is predominantly destined to store energy in the form of a single large lipid droplet, while brown adipose tissue exhibits multiple small lipid droplets, is rich in mitochondria, and is determined to dissipate energy as heat in the process on non-shivering thermogenesis. Consequently, BAT is found at strategic positions mostly in the anterior body, predisposed to protect the neck and chest regions of smaller animals or also human infants from hypothermia[1–3]. In addition, certain WAT depots, such as inguinal WAT, have the ability of browning, therefore referred to as inducible BAT, or beige or brite depots[3–6]. While the interscapular BAT depot, which is the most prominent BAT in adult mice, exists in human infants, it regresses during further growth. Nevertheless, thermogenic adipose tissue that can be stimulated by cold exposure has been identified in adult humans in the cervical, supraclavicular, axillary and paravertebral/aortic region, and its extent is inversely correlated to body-mass-index[1,7–11]. Given this role in energy expenditure through a thermogenic program, BAT became an attractive therapeutic target for obesity and associated metabolic syndromes[2,3,5,12–14].

An important aspect of understanding tissue function is its developmental ontogeny. Early lineage tracing analyses indicated that BAT depots primarily originate from descendants of the paraxial

[1]Institute for Chemistry and Biochemistry, Freie Universität Berlin, Thielallee 63, Berlin, Germany. [2]German Institute of Human Nutrition Potsdam-Rehbruecke, Arthur-Scheunert-Allee 114-116, 14558, Nuthetal, Germany. [3]Max Delbrück Center for Molecular Medicine in the Helmholtz Association, Hannoversche Str. 28, Berlin, Germany. [4]Berlin Institute of Health at Charité-Universitätsmedizin Berlin, Augustenburger Platz 1, Berlin, Germany. [5]Department of Hematology, Oncology and Tumor Immunology, Charité-Universitätsmedizin Berlin, Corporate Member of Freie Universität Berlin and Humboldt-Universität zu Berlin, Augustenburger Platz 1, Berlin, Germany. [6]Department of Cell Biology, Faculty of Medicine, Karl Landsteiner University, Dr. Karl-Dorrek-Str. 30, Krems, Austria. [7]Max Planck Institute for Molecular Genetics, Ihnestr. 63-73, Berlin, Germany. ✉e-mail: sigmar.stricker@fu-berlin.de; sigmar.stricker@kl.ac.at

mesoderm, which forms transient metameric structures called somites in early development. The somitic dermomyotome compartment is seen as the main source for all BAT depots, via cells from an En1[+], Myf5[+], or Pax7[+] lineage[15–17]. This appeared reasonable, as it linked BAT progenitors with myogenic progenitors, both forming highly metabolically active tissues[1]. However, recent analyses have increased the complexity of BAT origin. While the dermomyotomal Myf5[+] or Pax3[+] lineage contributed to the majority of interscapular and subscapular BAT, the cervical and peri-aortic depots contained significant amounts of adipocytes not showing a Myf5[+] or Pax3[+] origin, implying that subsets of BAT depots might derive from other unidentified lineages. Moreover, the dermomyotomal Myf5[+] / Pax3[+] lineage not only contributed to BAT, but also certain anterior WAT depots[18,19]. Subcutaneous WAT, on the other hand, was traced back to a lateral plate mesoderm lineage, which did not contribute to BAT depots[20,21]. This suggests that adipose tissue depots at different anatomical locations originate from different local progenitors, or combinations of such[12,22], a complexity that is still not fully understood.

Here we show that, in mouse embryos, the zinc finger transcription factor Odd skipped-related 1 (Osr1) is expressed in progenitors contributing to BAT and subcutaneous WAT across various anatomical locations. While on embryonic day 11.5 (E11.5), Osr1 is expressed in dermomyotome-derived BAT progenitors, consistent with the current understanding of BAT formation, we propose an additional population of Osr1[+] BAT and subcutaneous WAT progenitors located in the dorsal aortic compartment at day E9.5. Our data suggest that this population might represent the in vivo counterpart of mesoangioblasts, a multipotent transient progenitor cell type previously isolated from the dorsal aortic compartment[23] thereby revealing an alternative source of adipose tissue during development.

## Results

### Osr1 is expressed in adipogenic progenitors

Since Osr1 was expressed at E11.5 in lateral plate mesoderm-derived cells that in the limb areas contributed to fibroblasts and white adipocytes[24], we analyzed Osr1 expression using an enhanced GFP (eGFP) -CreERt2 (Osr1[GCE]) knock-in allele in nascent adipose tissue depots at other anatomical locations. This allele can be used to visualize the active Osr1 expression with GFP, or the lineage derived from Osr1-expressing cells via tamoxifen-mediated CreERt2 activation[25].

At E13.5, first accumulations of PPARγ[+] cells can be observed in the anterior trunk of the embryo on both sides of the neural tube at a dorso-medial position, where the medial part will later form the interscapular BAT (iBAT), and the lateral portions will form the subscapular BAT (sBAT) (Fig. 1a). At this developmental stage PPARγ[+] cells represent preadipocytes that have a spindle-shaped fibroblastic appearance and do not yet contain lipid droplets. At E14.5, the initial anlagen of iBAT and sBAT have separated (Fig. 1b), and lateral BAT (latBAT) and axillary WAT (axilWAT) can be identified (Fig. 1b), as can be cervical BAT (cBAT) (Fig. 1c).

In all depots analyzed, Osr1-eGFP expression was found at E13.5 and E14.5 mainly in PPARγ-negative cells surrounding the depots as well as in interstitial PPARγ-negative cells, including vessel-associated cells (Fig.1a–c and Supplementary Fig. 1a). Intriguingly, cells surrounding the presumptive iBAT showed weaker Osr1-eGFP expression compared to the other depots (Fig. 1a, b). Both, the periphery of adipose depots as well as the vessel niche, are known to harbor adipogenic progenitors that mediate adipose tissue growth[26–29], suggesting Osr1 expression in the adipogenic progenitor pools associated with these depots. In addition, Osr1-eGFP expression was found in PPARγ[+] preadipocytes predominantly at the periphery of the depots (Fig. 1a, b and Supplementary Fig. 1a), consistent with the assumption that these cells may originate from PPARγ-negative/Osr1[+] progenitors in the periphery. Again, in the nascent interscapular BAT (iBAT) depot, only a

few Osr1-eGFP/PPARγ[+] preadipocytes were observed (Fig. 1a, b). At E18.5, when lipid-loaded adipocytes are identifiable, Osr1-eGFP expression can be found in vessel-associated cells, but not in mature adipocytes (Fig. 1d), suggesting downregulation of Osr1 expression during overt adipogenic differentiation and continuous expression in the vascular niche.

We confirmed Osr1 expression in adipogenic progenitors and preadipocytes using a published single-cell dataset of developing postnatal day 3 (p3) and adult perivascular adipose tissue[27]. In both datasets, Osr1 was detectable in smooth muscle cells (SMC) residing in the aortic adventitial region, fibroblastic cells (annotated as intermediate cells in this dataset), as well as in adipogenic progenitor cells and preadipocytes. Osr1 expression was not detected in mature adipocytes (Supplementary Fig. 1b). This collectively suggests that Osr1 is expressed in progenitors and preadipocytes of WAT and BAT at E13.5 / 14.5 in the anterior trunk of the embryo.

To analyze whether Osr1 is expressed in earlier populations of adipogenic progenitors present before the emergence adipose depots, we performed genetic lineage tracing via crossing Osr1[GCE/+] mice to Rosa26[mTmG] reporter mice[30], hereafter called R[mTmG]. We first pulsed tamoxifen at E11.5, a stage where no accumulations of PPARγ[+] cells at sites of prospective AT formation can be observed yet[31]. Analysis at E18.5, when initial formation of adipose depots is thought to be completed[32], revealed E11.5 Osr1 cell descendants in sBAT and cBAT, but scarcely in iBAT (Fig. 1e) and in latBAT, axiWAT and subWAT (Supplementary Fig. 1c; overview of adipose depots labeled by FABP4 in the E18.5 embryo shown in Supplementary Fig. 1d). Quantification of Osr1 descendants showed approx. 18% and 12% contribution for sBAT and cBAT, respectively, while contribution to iBAT was negligible (Fig. 1e). This suggests that the Osr1[+] lineage has different relevance for distinct BAT depots. We confirmed the identity of E11.5 Osr1[+] lineage traced cells in the E18.5 sBAT depot as definitive brown adipocytes by labeling for Perilipin and UCP1 (Supplementary Fig. 1e).

The overall modest levels of Osr1 lineage contribution to sBAT and cBAT may be explained by the low efficacy of the Osr1-CreERt2 allele, as we have observed before[24,33], thus pulse labeling may only partially capture the full population. Furthermore, adipogenesis is a dynamic process continuously recruiting progenitor cells over an extended period; Osr1 may be expressed only transiently in these progenitors, implying that tamoxifen administration at one specific time point only leads to labeling of a fraction of progenitors.

We next analyzed the long-term contribution of Osr1[+] progenitors to adipose tissues, including depots that form later, as aortic BAT or epididymal WAT. To be able to identify Osr1 lineage descendants after birth, two consecutive doses of tamoxifen are required[33]. We pulse-labeled at E11.5 + E12.5 and analyzed at 11 weeks of age. Due to high mortality of tamoxifen-pulsed fetuses after birth, we were only able to retrieve 2 biological replicates, thus data were not quantified. This nevertheless indicated persistent contribution of Osr1 descendants to sBAT, cBAT and ingWAT, but also to adult aortic BAT (aBAT, which is not detectable in the embryo) as well as to adult iBAT (Supplementary Fig. 2a). No contribution of E11.5/12.5 Osr1[+] cells was observed in epididymal WAT (eWAT) (Supplementary Fig. 2a), consistent with the emergence of this depot only after birth[12,34].

Postnatal adipose tissue underlies constant turnover by associated fibroblast-like adipogenic progenitor cells in adventitial cell layers or the stromal vascular niche[28,29,35], we thus analyzed whether developmental Osr1[+] cells contributed to this progenitor pool. Indeed, tamoxifen labeling at E11.5 + E12.5 showed contribution to smooth muscle actin-positive perivascular cells in sBAT, cBAT, iBAT, ingWAT and aBAT (Supplementary Fig. 2b).

In summary, Osr1 is expressed in early embryonal adipogenic progenitors at E11.5 that contribute to a broad range of BAT depots and (inguinal) WAT.

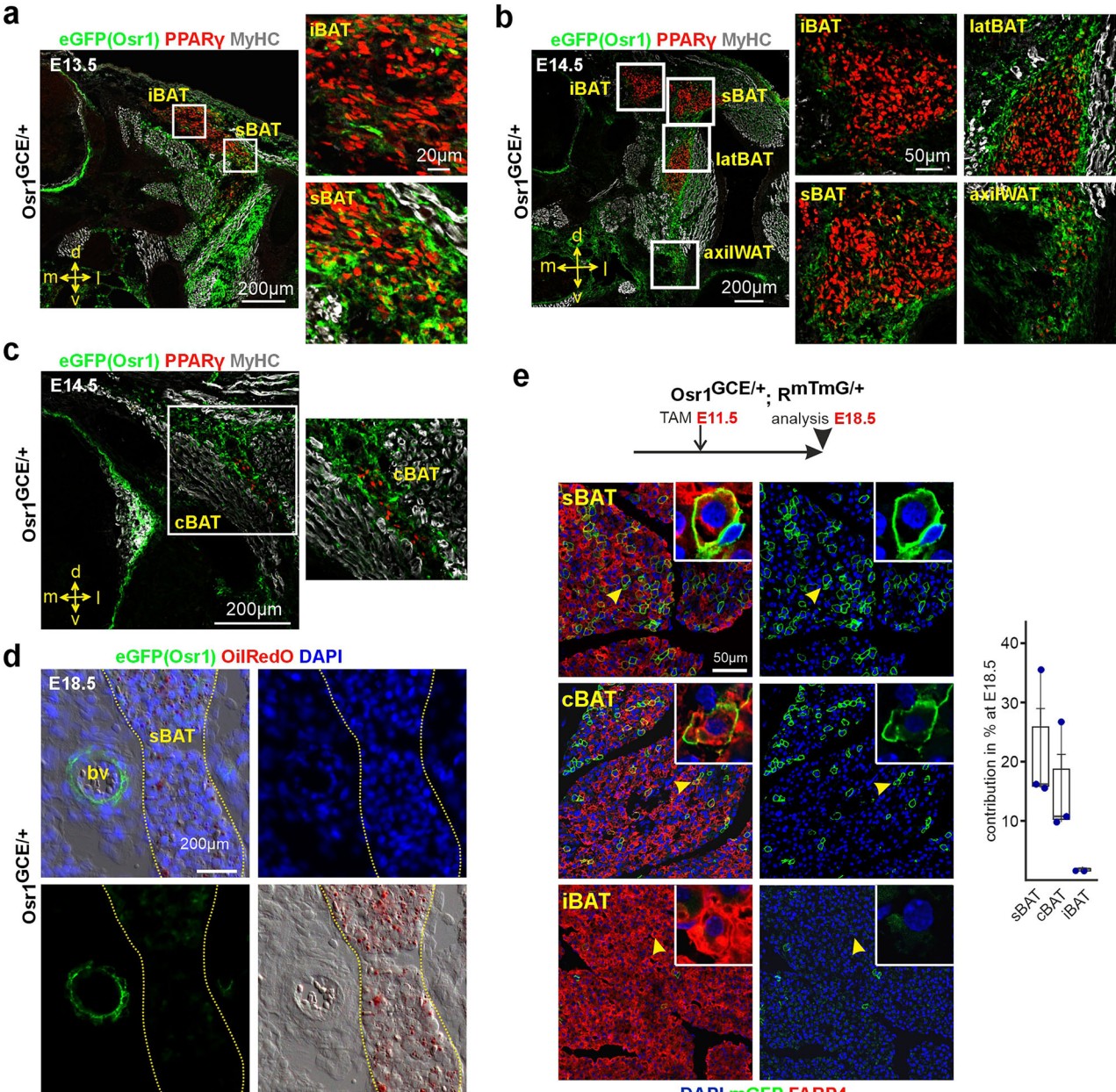

**Fig. 1 | Osr1 is expressed in adipogenic progenitors and preadipocyes.**
**a–c** Immunolabeling of thoracic level sections of E13.5 (representative image of
*n* = 2) and E14.5 (representative images of *n* = 4) Osr1$^{GCE/+}$ embryos, respectively, for
eGFP (Osr1 reporter; green), PPARγ (preadipocytes; red) and MyHC (muscle; white);
iBAT: interscapular BAT; sBAT: subscapular BAT; latBAT: lateral BAT; axilWAT:
axillary WAT; cBAT: cervical BAT. Boxed regions are shown as magnifications right.
Orientation is indicated: d: dorsal; l: lateral; m: medial; v: ventral. **d** Labeling of E18.5
Osr1$^{GCE/+}$ thoracic sections (sBAT area shown; representative image of *n* = 3) for
eGFP (Osr1 reporter, green), Oil red O (lipids, red) and DAPI (nuclei; blue). sBAT
depot indicated by dashed lines. **e** Genetic lineage tracing of E11.5 Osr1⁺ cells to
E18.5 (schematic depiction top) using Osr1$^{GCE/+}$;R$^{mTmG/+}$ embryos; representative
images of n = 3. Indicated BAT depots are shown, tissue sections were labeled for
mGFP (Osr1 lineage; green), FABP4 (adipocytes; red) and DAPI (nuclei; blue).
Arrowheads indicate cells shown in magnification inserts. Box plot quantification

shown right (Averages of *n* = 3 biological replicates shown as dots, error bar depicts
standard error of means; each embryo was investigated on multiple sections). The
sBAT box plot shows a range of values, a minimum of 15,53 %, a 1st quartile (lower
boundary = 25th percentile) of 15,86%, an interquartile range (IQR) of 6,36%, a
median of 16,19%, a mean of 22,41%, a standard error of the mean (SEM) of 6,56%, a
maximum of 35,52% and a 3rd quartile (upper boundary = 75th percentile) of
25,86%. The cBAT box plot displays a minimum of 9,79%, a 1st quartile of 10,27%, an
IQR of 8,46%, a median of 10,75%, a mean of 15,74%, a SEM of 5,48%, a maximum of
26,70% and a 3rd quartile of 18,73%. The iBAT box plot displays a minimum of 1,55%,
a 1st quartile of 1,55%, an IQR of 0,37%, a median of 1,55%, a mean of 1,80%, a SEM of
0,25%, a maximum of 2,29% and a 3rd quartile of 1,92%. In general, the whiskers
represent the distance between the minimum and the maximum. Source data are
provided as a Source Data file.

### Distinct BAT progenitors express Osr1 at E9.5 and E11.5

Osr1 expression is broad and occurs in unrelated tissue types such as
connective tissues, lungs and kidneys[25,36,37], thus Osr1 expression in the
embryo cannot solely pinpoint adipogenic progenitors. Since BAT was
previously shown to derive from the dermomyotomal lineage cells

having expressed *Pax7* or *Myf5*[15,16,31], we asked if Osr1-eGFP expression
could be detected in these early lineages to identify the Osr1⁺ subset
that contributes to BAT. We found Osr1-eGFP expression at E11.5
dorso-lateral of the dermomyotome, in the central dermomyotome
region intermingling with PAX7⁺ cells, and ventro-medial of the

dermomyotome at the border of the abdominal wall (Fig. 2a). Here, we also observed co-expression of Osr1-eGFP and PPARγ (Fig. 2a, yellow arrows), indicating early Osr1/ PPARγ⁺ preadipocytes arising in this region. EBF2 was proposed to mark the earliest dermomyotomal pre-BAT progenitors[31], co-expression of EBF2 and OSR1 proteins in the central dermomyotome area (Fig. 2b, yellow arrow) indicated a pre-BAT identity of Osr1⁺ cells in this region.

XGal staining detecting expression of an Osr1-LacZ knock-in allele[38] was observed in Pax7 lineage descendants in the dermomyotome area at E11.5 (Fig. 2c, top row). By E12.5, Osr1-LacZ signal was found in Pax7 and Myf5 lineage descendants that had extended laterally and ventrally towards and into presumptive sBAT formation sites (Fig. 2c, middle and bottom rows). Consistent with low Osr1-lineage contribution to iBAT (Fig. 1e), Osr1-LacZ expression at this stage was predominantly restricted to Myf5 descendants localized in the presumptive sBAT region, with minimal expression in the dorsal iBAT domain (Fig. 2c bottom row).

This altogether indicates that at E11.5, Osr1 is expressed in EBF2⁺ adipose progenitor cells derived from the dermomyotome that later accumulate in and around areas of sBAT and cBAT location.

The contribution of Pax7⁺ dermomyotome cells to BAT depots can be traced back to E9.5[15]. However, we observed no expression of Osr1-reporter or OSR1 protein expression in the dermomyotome or dermomyotome derivates at E9.5. OSR1 protein expression was not noticeable in or close to PAX7⁺ cells (Fig. 3a and Supplementary Fig. 3a). In line, Osr1-LacZ reporter expression was not detected in the dermomyotomal region at E9.5 (Supplementary Fig. 3b). We also excluded the possibility of Osr1 expression in cells that emanated from the Myf5⁺ paraxial mesoderm prior to E9.5, as no expression of Osr1-LacZ was identified in Myf5-constitutive-Cre lineage traced cells at E9.5 (Supplementary Fig. 3b). OSR1⁺ cells were only found in the center of the embryo, labeling cells of the lung buds, the foregut, and in loose mesenchymal cells between the dorsal aorta and the foregut in the anterior embryo representing the aorta-gonad-mesonephros region (AGM) (Fig. 3a and Supplementary Fig. 3a, b). This was confirmed by analyzing Osr1 expression in the MOSTA spatial transcriptomics atlas[39] showing expression in the trunk predominantly in the mesenchyme of the anterior AGM and the associated liver primordium (Fig. 3b). In the posterior embryo, OSR1 expression was found in the intermediate mesoderm and gut region (Supplementary Fig. 3a) in line with previous observations[25,37]. No expression of OSR1 was found in the limb buds at E9.5 (Supplementary Fig. 3a).

Nevertheless, lineage tracing of E9.5 Osr1⁺ cells uncovered a contribution to E18.5 sBAT and cBAT at comparable or even slightly higher levels compared to tracing E11.5 Osr1⁺ cells (Fig. 3c). E9.5 Osr1 lineage contribution was also seen in latBAT and in anterior white adipose depots, axiWAT and subWAT (Supplementary Fig. 3c). Low contribution of E9.5 Osr1⁺ cells was observed in iBAT (Fig. 3c). Labeling for Perilipin and UCP1 (Supplementary Fig. 3d) again confirmed bona-fide BAT identity for Osr1-lineage traced cells.

We then compared the timing of E9.5 vs. E11.5 Osr1-lineage contribution to sBAT by analysis at E14.5. E9.5 tamoxifen administration revealed only a low contribution to PPARγ⁺ cells in the nascent sBAT depot at E14.5, which were mainly located in the ventral portion (Fig. 3d). In contrast, E11.5 tamoxifen administration resulted in significant labeling of PPARγ⁺ cells in ventral and dorsal regions of the depot (Fig. 3d).

The low contribution of E9.5 cells to the nascent E14.5 sBAT could be the result of low Osr1-CreERt2 activity within dermomyotomal cells that express Osr1 at sub-detection levels; however, the comparable contribution levels of E9.5 and E11.5 pulsing seen at E18.5 argue against this assumption. Alternatively, the adipogenic population labeled when treating Osr1^GCE/R^mTmG embryos at E9.5 might represent a lineage derived from cells that express high levels of Osr1 in the anterior embryo at E9.5, such as those in the lung, AGM and foregut/esophagus

regions (Fig. 3a, b and Supplementary Fig. 3a, b). According to this hypothesis, an alternative, early mesenchymal source distinct from the dermomyotome may contribute to anterior BAT and WAT depots during prenatal development.

## Single-cell RNA sequencing of E9.5 and E11.5 Osr1⁺ cells

To gain deeper insight into the Osr1⁺ cell pools at E9.5 and E11.5, we performed single-cell RNA sequencing of Osr1-eGFP cells isolated by FACS (Supplementary Fig. 4) from Osr1^GCE embryos at E9.5 or E11.5. Clustering by Uniform Manifold Approximation and Projection (UMAP) identified 12 overlapping clusters (cl0 – cl11) for both time points (Fig. 4a). The three centrally located clusters 0, 1 and 2 represented the most abundant cell types at both stages (Fig. 4a) and are therefore considered main clusters. As a quality check, we mapped our E9.5 and E11.5 Osr1⁺ cell pools to the whole embryo dataset from Cao et al.[40] (Supplementary Fig. 5a). Our dataset at both time points recapitulates the distribution of Osr1⁺ cells in this reference dataset to a high extent, reassuring that we captured the full Osr1⁺ cell population.

Cluster annotation of the integrated E9.5 and E11.5 datasets (Fig. 4b) suggested various cell type identities for the smaller clusters; cl3: neurogenic; cl4 epithelial; cl5: meningeal fibroblasts; cl6: endothelial; cl7: cardiomyocytes; cl8: primitive erythrocytes; cl9: megakaryocytes; cl10: macrophages and cl11: hepatocytes, all expressing representative marker genes (Supplementary Fig. 5b); these clusters were not used for further analysis. Clusters 0 and 2 have a high similarity among their gene signatures. Both show high expression of Pdgfra, Prrx1, Lum and Col3a1, and the highest expression levels of Osr1 itself (Supplementary Fig. 5b), altogether indicating a mesenchymal stromal cell identity. In addition, cl2 was enriched in G2/M phase cell cycle genes like Cdk1 and Ccnb1 (Supplementary Fig. 5b); consequently, we consider cl2 as the proliferative counterpart to cl0. Cluster 1 showed top enrichment for mt-Nd1, Nckap5, Slit2, Slit3, and Fbn2 (Supplementary Fig. 5b), not uniquely defining a clear identity. Amongst the enriched genes in this cluster, but not exclusive for it, we found Pdgfra and Prrx1, suggesting a stromal identity (Supplementary Fig. 5b, c), Acta2 marking smooth muscle, Acta1, Myh4 and Myh7 typical for skeletal and cardiac myogenic cells, and Cdh5 and Kdr marking endothelial cells (Supplementary Fig. 5c), indicating a mixed mesenchymal identity. This altogether suggests that at E9.5 and E11.5, the majority of Osr1⁺ cells consist of a variety of mesenchymal stromal cell types. Main cl1 as well as small clusters appeared to decrease between E9.5 and E11.5, while clusters 0 and 2 increased (Fig. 4a), suggesting a consolidation of Osr1⁺ cells to a common stromal / connective tissue cell trajectory during development.

We next aimed to identify brown adipogenic progenitors individually in both datasets. We used the top-expressed genes of a Myf5/Ebf2⁺ BAT progenitor population[31] (gene list shown in Supplementary Fig. 5d) as a pre-BAT signature and conducted module scoring on the Osr1 datasets. The pre-BAT signature was identified mainly in cl0 and weaker in cl1 and cl2 in the E11.5 dataset, whereas at E9.5 it matched only weakly with cl1 (Fig. 4c and Supplementary Fig. 5e). Ebf2 expression itself showed a similar distribution, predominating in cl0 at E11.5 (Fig. 4c) and overlapping the pre-BAT signature in a subset of cl0 cells (Supplementary Fig. 5e), suggesting that these cells reflect the OSR1/EBF2⁺ cells identified via immunolabeling (Fig. 2b). Ebf2 expression was weaker in the E9.5 dataset (Fig. 4c and Supplementary Fig. 5e), in accordance with Wang et al.[31] who did not identify EBF2 expression before E11.5.

To further validate the putative adipogenic signatures of Osr1⁺ cells, we selected the top 50 signature genes of progenitor, intermediate and preadipocyte cells, respectively, from perivascular BAT of the P3 thoracic aorta[27], and performed module scoring to investigate how these signatures distributed in our E9.5 and E11.5 cl0, cl1 and cl2. Confirming the Ebf2 data, signatures mainly matched E11.5 cl0 and to a low degree, E9.5 cl1 (Fig. 4d).

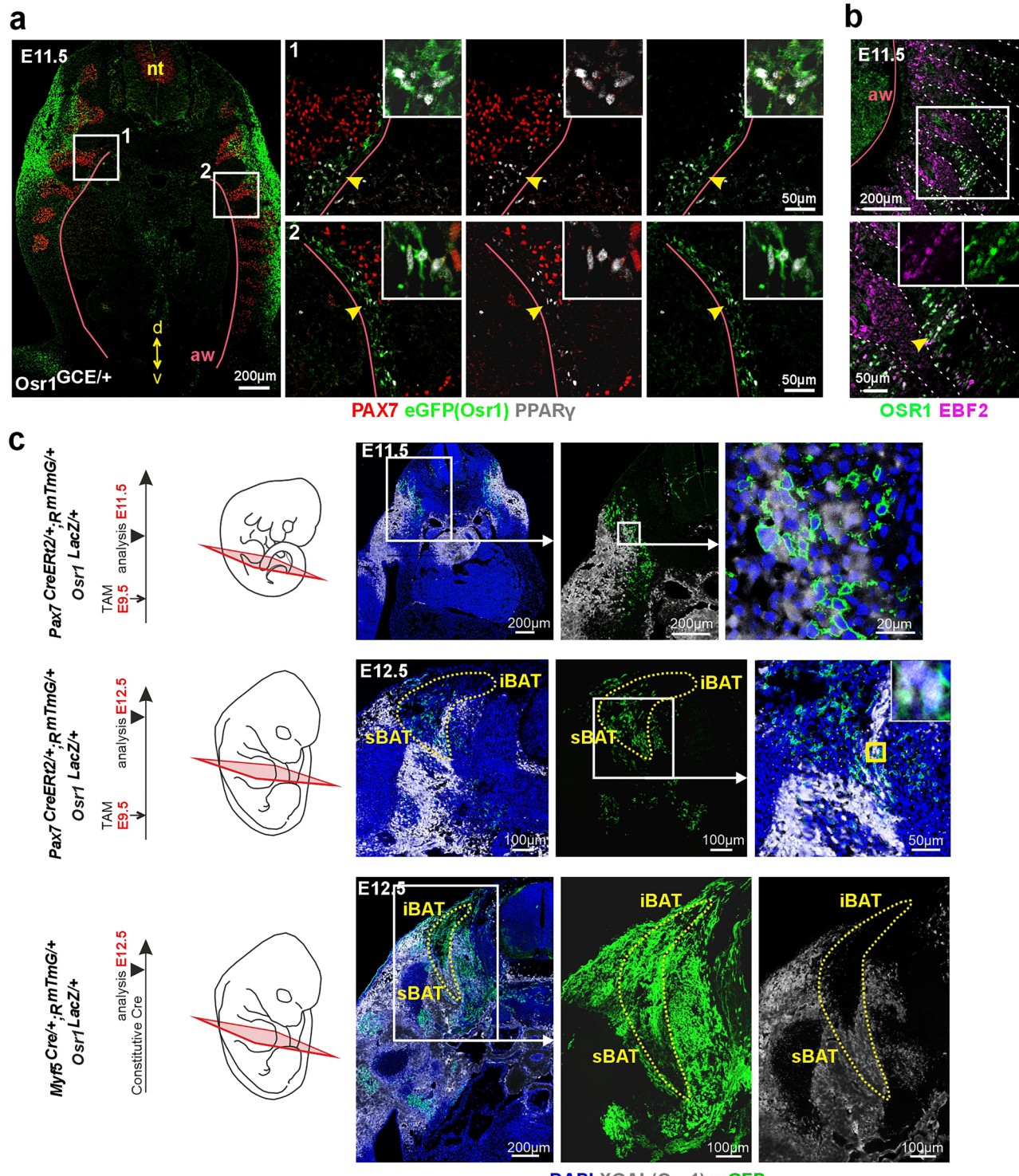

**Fig. 2 | E11.5 Osr1⁺ cells comprise dermomyotomal adipogenic progenitors.**
**a** Cross-sections of E11.5 Osr1$^{GCE/+}$ embryos at thoracic level labeled for eGFP (Osr1 reporter; green), PAX7 (red) and PPARγ (preadipocytes; white). Cells coexpressing eGFP (Osr1) and PPARγ indicated by arrowheads, see magnification inserts. Abdominal wall (aw) indicated by magenta line; nt: neural tube. Orientation is indicated: d: dorsal; v: ventral; representative image of $n = 4$. **b** Parasagittal sections of E11.5 Osr1$^{GCE/+}$ embryos at thoracic level labeled for OSR1 (green) and EBF2 (magenta). Area of OSR1 and EBF2 coexpression is indicated by an arrowhead, see magnification inserts. Areas of the dermomyotomes are indicated by dashed lines,

the abdominal wall (aw) indicated by a purple line; representative image of $n = 3$. **c** Genetic lineage tracing of Pax7⁺ cells (top and middle rows; representative images of $n = 2$ for each condition), and Myf5⁺ cells (bottom row; representative image of $n = 4$). Pax7CreERt2 was induced by tamoxifen administration at E9.5, Myf5Cre is constitutive. Analysis at E11.5 (top row) or E12.5 (middle and bottom rows); see schematic depictions to the left. Tissue sections were labeled for mGFP (Pax7 or Myf5 lineage, green), XGAL (*Osr1* expression from Osr1-LacZ reporter allele, white) and DAPI (nuclei, blue).

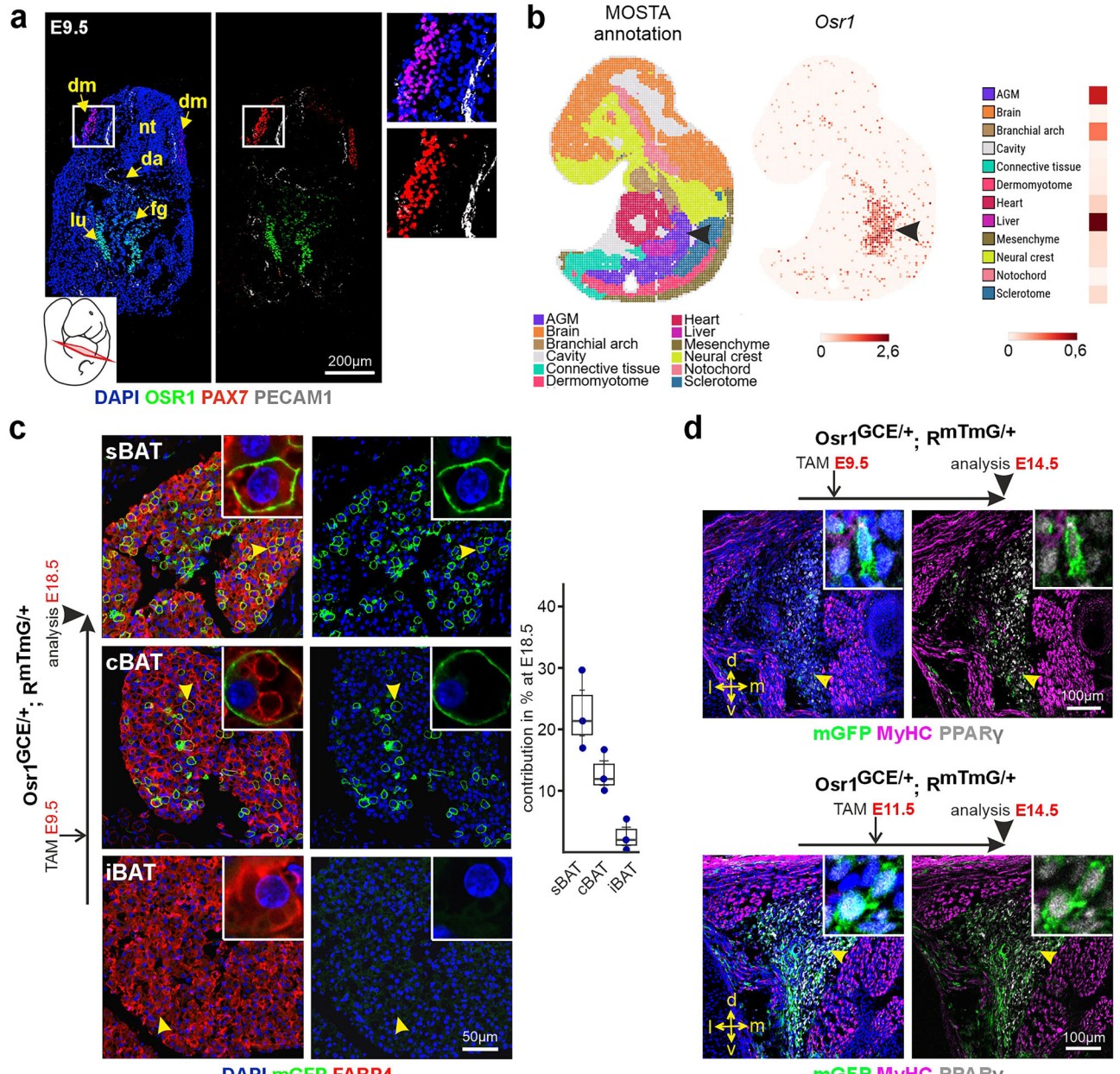

**Fig. 3 | At E9.5, Osr1 labels distinct, non-dermomyotomal BAT progenitors.**
**a** Section of E9.5 wild type mouse embryo at the level as indicated, labeled for OSR1 (green), PAX7 (red), PECAM1 (white) and DAPI (nuclei; blue). Boxed area shown as magnification right. Abbreviations: *da* dorsal aorta; *dm* dermomyotome; *fg*: foregut; *lu* lung bud; *nt* neural tube; representative image of $n = 3$. **b** Mapping of *Osr1* expression on an E9.5 embryo section of the MOSTA atlas; cell type annotation shown left, Osr1 spatial expression and heatmap depiction of expression shown right. The aorta-gonad-mesonephros region (AGM) is pointed out by an arrowhead. **c** Genetic lineage tracing of E9.5 Osr1+ cells to E18.5 (schematic depiction left) using Osr1GCE/+;RmTmG/+ embryos; representative images of $n = 3$. Indicated BAT depots are shown, tissue sections were labeled for mGFP (Osr1 lineage; green), FABP4 (adipocytes; red) and DAPI (nuclei; blue). Arrowheads indicate cells shown in magnification inserts. Box plot quantification shown right (Averages of $n = 3$ biological replicates shown as dots, error bar depicts standard error of means; each embryo was investigated on multiple sections). The sBAT box plot shows a range of values, a minimum of 16,96 %, a 1st quartile (lower boundary = 25th percentile) of 19,16%, an

interquartile range (IQR) of 6,36%, a median of 21,37%, a mean of 22,67%, a standard error of the mean (SEM) of 3,73% , a maximum of 29,67% and a 3rd quartile (upper boundary = 75th percentile) of 25,52%. The cBAT box plot displays a minimum of 10,05%, a 1st quartile of 10,99%, an IQR of 3,34%, a median of 11,93%, a mean of 12,90%, a SEM of 1,98%, a maximum of 16,72% and a 3rd quartile of 14,32%. The iBAT box plot displays a minimum of 0,33%, a 1st quartile of 1,17%, an IQR of 2,55%, a median of 2,01%, a mean of 2,59%, a SEM of 1,50%, a maximum of 5,42% and a 33rd quartile of 3,72%. In general, the whiskers represent the distance between the minimum and the maximum. **d** Genetic lineage tracing of E9.5 (top; representative image of $n = 4$) or E11.5 (bottom; representative image of $n = 5$) Osr1+ cells to E14.5 (schematic depiction above images) using Osr1GCE/+;RmTmG/+ embryos. Area of sBAT shown, tissue sections were labeled for mGFP (Osr1 lineage; green), PPARγ (pre-adipocytes; white), MyHC (muscle; magenta) and DAPI (nuclei; blue). Arrowheads indicate cells shown in magnification inserts. Orientation is indicated: d: dorsal; l: lateral; m: medial; v: ventral. Source data are provided as a Source Data file.

Furthermore, we extracted the cells that were positive for the pre-BAT signature from our E11.5 dataset and deconvoluted their marker profile (top 50 genes) onto E11.5 sections of the MOSTA atlas. This confirmed presence of Osr1-pre-BAT cells predominantly in the region

of the anterior dermomyotome and associated connective tissue like cells (arrowheads, Fig. 4e). The Osr1-pre-BAT cell signature strongly overlapped with the anterior expression domain of *Ebf2* and a confined expression domain of *Osr1* on the same MOSTA section (Fig. 4e),

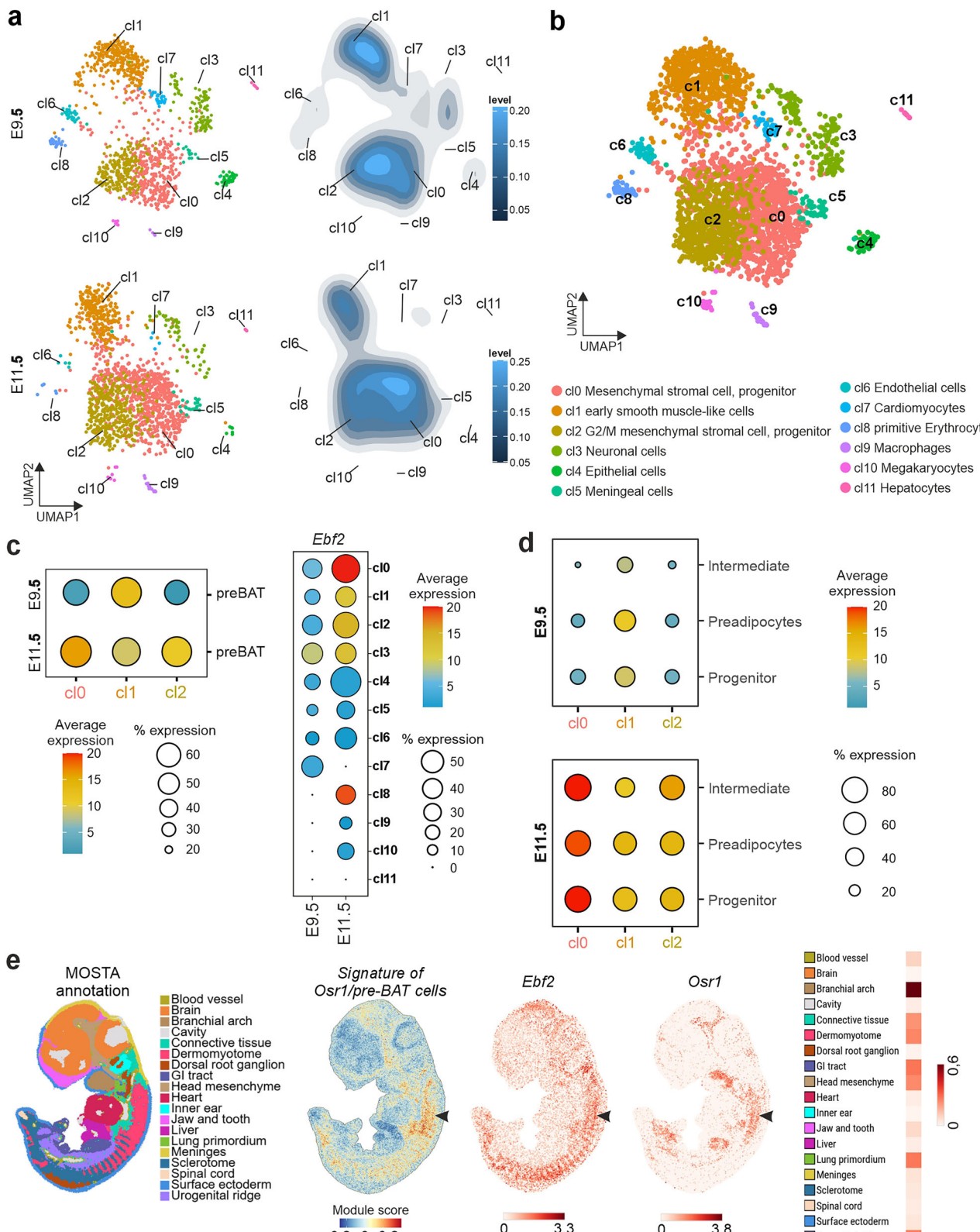

**Fig. 4 | Single-cell analysis of Osr1+ cells at E9.5 and E11.5. a** UMAP visualization of E9.5 (top) and E11.5 (bottom) Osr1-eGFP single-cell sequencing analysis. Density plots shown right. **b** UMAP visualization and cluster identities of combined E9.5 and E11.5 Osr1-eGFP cells. **c** Dot plot depiction of pre-BAT signature (left) and *Ebf2* expression (right) mapped to E9.5 or E11.5 Osr1-eGFP single-cell data. **d** Dot plot depiction of P3 peri-aortic adipogenic progenitor signatures from Angueira et al. 2021, mapped to E9.5 or E11.5 Osr1-eGFP single-cell data. **e** Deconvolution analysis of Osr1/preBAT cell signatures on the MOSTA atlas at E11.5 (annotation shown left; arrowheads point out the anterior dermomyotome region). Expression of *Ebf2* and *Osr1* are shown for comparison; right side: heatmap depiction of Osr1 expression according to MOSTA annotation.

altogether corroborating immunostaining data (Fig. 2a, b). At E12.5, there was still widespread overlap of *Osr1* and *Ebf2* expression in anterior-dorsal regions mainly annotated as connective tissue (Supplementary Fig. 6a). At E14.5, *Ppary* expression was found at the presumptive site of BAT formation, here, *Ebf2* expression was found overlapping *Ppary*, while *Osr1* expression was preferentially seen in the circumference of *Ppary expression* (Supplementary Fig. 6a), corroborating immunostaining data (Fig. 1b). Intriguingly, this expression domain overlaps with a pattern recently described by Jun et al.[41] for a signature of presumptive BAT progenitors characterized by expression of *Dpp4*. Of note, *Dpp4* expression is not detectable before E13.5[41] and therefore not present in our E9.5 and E11.5 datasets (Supplementary Fig. 6b).

Altogether, these data indicate that Osr1+ cells with a pre-adipogenic signature anatomically mapping to the anterior dermo-myotome via EBF2 coexpression are identified at E11.5 in cl0, whereas this signature does not clearly emerge in the E9.5 dataset. A relative match of the pre-adipogenic signature with E9.5 cl1, along with its expression of conflicting cell marker genes, suggests that these cells may have probable multipotent characteristics.

### An E9.5 Osr1+ multipotent progenitor population

We therefore further analyzed cl1 in comparison to cl0 and examined the evolution of the corresponding gene sets/signatures between E9.5 and E11.5. While cardiac muscle markers *Myl7*, *Tnnt2*, the smooth muscle markers *Acta2* and *Tagln*[42], and endothelial markers *Kdr* and *Tek* all were expressed at high levels in cl1 at E9.5, those genes were downregulated at E11.5. In contrast, *Col1a2* and *Col3a1* increased in cl1 at E11.5 (Fig. 5a), suggesting the loss of a myogenic and angiogenic identity at the expense of a stromal connective tissue-like identity. Likewise, Enrichr analysis indicated a mixed identity showing e.g., myogenic and vascular endothelial signatures for E9.5 cl1, while E11.5 cl1 showed predominant stromal cell and smooth muscle-like identity (Fig. 5b).

We also noticed a shift in gene expression in cl0 between E9.5 and E11.5. While the E9.5 cl0 showed smooth muscle gene expression in addition to stromal cell markers, at E11.5 expression of the smooth muscle genes decreased whereas expression of stromal markers increased (Fig. 5a). Gene ontology analysis via Enrichr confirmed predominant stromal identity of cl0 throughout, with additional detection of smooth muscle signatures at E9.5 (Fig. 5b). Altogether, this is in line with the idea that E9.5 Osr1+ cells match less with a clear definition of cell identity or cell lineage compared to mostly stromal-like Osr1+ cells at E11.5 and also suggests that E9.5 cl0 and cl1 are more closely related than the respective E11.5 clusters.

A close relationship between cl1 and cl0 at E9.5, but not E11.5, was confirmed by Metascape circos overlap analysis, which confirms that at E9.5 cl1 shares more signature genes with cl0 and cl2 than at E11.5 (Supplementary Fig. 7a). Using Velocity analysis, we observed a directional flow from cl1 to cl0/cl2 specifically at E9.5, suggesting a possible lineage connection (Fig. 5c). The preferential direction of vectors was not observed at E11.5 where we instead observed vectors with multiple directions, possibly reflecting intrinsic fluctuations in gene expression (Fig. 5c). A direct lineage relationship between E9.5 cl1 and cl0 was further supported by developmental trajectory reconstruction analysis using the STREAM tool, which identified a direct transition between cl1 and cl0/cl2 in the E9.5 dataset (Fig. 5d; underlying gene expression changes represented in pseudotime heatmap in Supplementary Fig. 7b). Again, this relationship was not detected at E11.5 (Fig. 5d). Molecularly, the temporary transition from cl1 to cl0/2 at E9.5 as identified by STREAM analysis is characterized by the loss of expression of endothelial markers *Kdr* and *Tek*, endothelial and stem cell marker *Cd34*, skeletal / cardiac myogenic markers *Acta1* and *Actc1*, and smooth muscle marker *Tagln*, and the induction of mesenchymal stromal markers as *Pdgfra* and *Meox1* (Fig. 5e).

To further investigate the lineage identities in our Osr1 dataset, we used embryonic mouse limb single-cell sequencing data[43] and derived myogenic, tenogenic and chondrogenic module score signatures. While the myogenic signature exclusively mapped to E9.5 cl1 in line with the analysis above, tenogenic and chondrogenic signatures mapped predominantly to cl1 at E9.5 and shifted to cl0/2 at E11.5 (Supplementary Fig. 7c) in line with a hypothesized flow of cells in this direction. Expression of key marker genes that describe a tenogenic (*Scx*) as well as osteo-chondrogenic (*Sox9* and *Runx2*) identity were detected in an overlapping pattern (Supplementary Fig. 7d). Intriguingly, we have shown before that E13.5 Osr1+ cells still have expression of alternative lineage genes as *Scx* and *Runx2*, but that chondrogenic, tenogenic and also myogenic differentiation is actively repressed by Osr1 itself[24,44]; indeed, Osr1 is a direct repressor of Sox9[45]. Comparing Osr1 expression levels between the E9.5 and E11.5 datasets suggested an upregulation of Osr1 over time as cells progress to an overall stromal fate (Supplementary Fig. 7e). This would agree with the assumption that a lower level of Osr1 expression in Osr1+ cells at E9.5 may be instrumental to maintain a multipotent fate, which becomes more and more restricted as Osr1 expression increases.

This altogether suggests that the E9.5 cl1 may hold cells of a promiscuous identity, indicating a multilineage potential that could possibly contribute to different mesenchymal lineages. Indeed, in line with the predictions from the single-cell sequencing data, E9.5 Osr1+ cells were multipotent in vivo. While E11.5 Osr1+ cells are strongly lineage restricted, exclusively giving rise to adipocytes and connective tissue fibroblasts[24,33], Tamoxifen induction at E9.5 showed Osr1-lineage contribution, in addition to connective tissue and adipocytes, to skeletal muscle, skeletal tissues and blood vessel walls at E18.5 in the mediolateral part of the trunk (Fig. 5f).

This altogether confirms the multipotent capacities of the E9.5 Osr1+ population and finally raises the questions where these multipotent Osr1+ progenitors may be located in the embryo.

### An E9.5 migratory mesoangioblast-like Osr1+ cell population

The properties of E9.5 cl1 – the expression of a promiscuous set of marker genes for smooth, skeletal and cardiac muscle in addition to endothelial markers, alongside the capacity for multilineage differentiation, resemble those of so-called mesoangioblasts (MAB). MABs were initially isolated from the AGM, or more specifically from the isolated dorsal aorta compartment (DAC) thereof, of mouse embryos around E9.5 and E10.5[23,46]. Embryonic MABs were characterized as cells that in vitro, depending on specific culture conditions or co-culture with other cells, differentiate to multiple lineages such as cardiac or skeletal muscle, smooth muscle, endothelial cells, and also adipocytes. Their in vitro profile includes endothelial markers such as *Kdr*, *Cd31* (encoding PECAM1) and *Cd34,* as well as myogenic markers as *Acta2*[23,46,47], suggesting a potential relation with our E9.5 cl1 Osr1+ cells. In sections of E9.5 mouse embryos showing the dorsal aorta at the thoracic level, we observed an accumulation of OSR1+ cells in the anterior DAC (Fig. 6a). Intriguingly, co-expression of OSR1 and PECAM1 was exclusively observed in cells located in this anatomical location (Fig. 6a). This raises the hypothesis that E9.5 cl1 Osr1+ cells may comprise MAB-like cells. To explore this possibility, we performed module scoring to analyze the representation of an MAB signature on our Osr1-eGFP single-cell dataset[47,48]. As the MAB signature (Supplementary Fig. 8a) contains endothelial markers, it unsurprisingly mapped to cl6 in both datasets (Fig. 6b). In addition, the MAB signature also highlighted with cl1 at E9.5, but not at E11.5 (Fig. 6b). Individual markers for both endothelium and MABs, *Kdr* and *Cd34*, recapitulated this pattern (Supplementary Fig. 8b), as did *Acta2*, and also *Emcn* and *Aplnr* (Supplementary Fig. 8b), which were identified as markers for embryonic stem cell-derived MABs or MAB precursors, respectively[49]. These data are consistent with the possibility that cells in the E9.5 cl1 may be endowed with MAB-like properties.

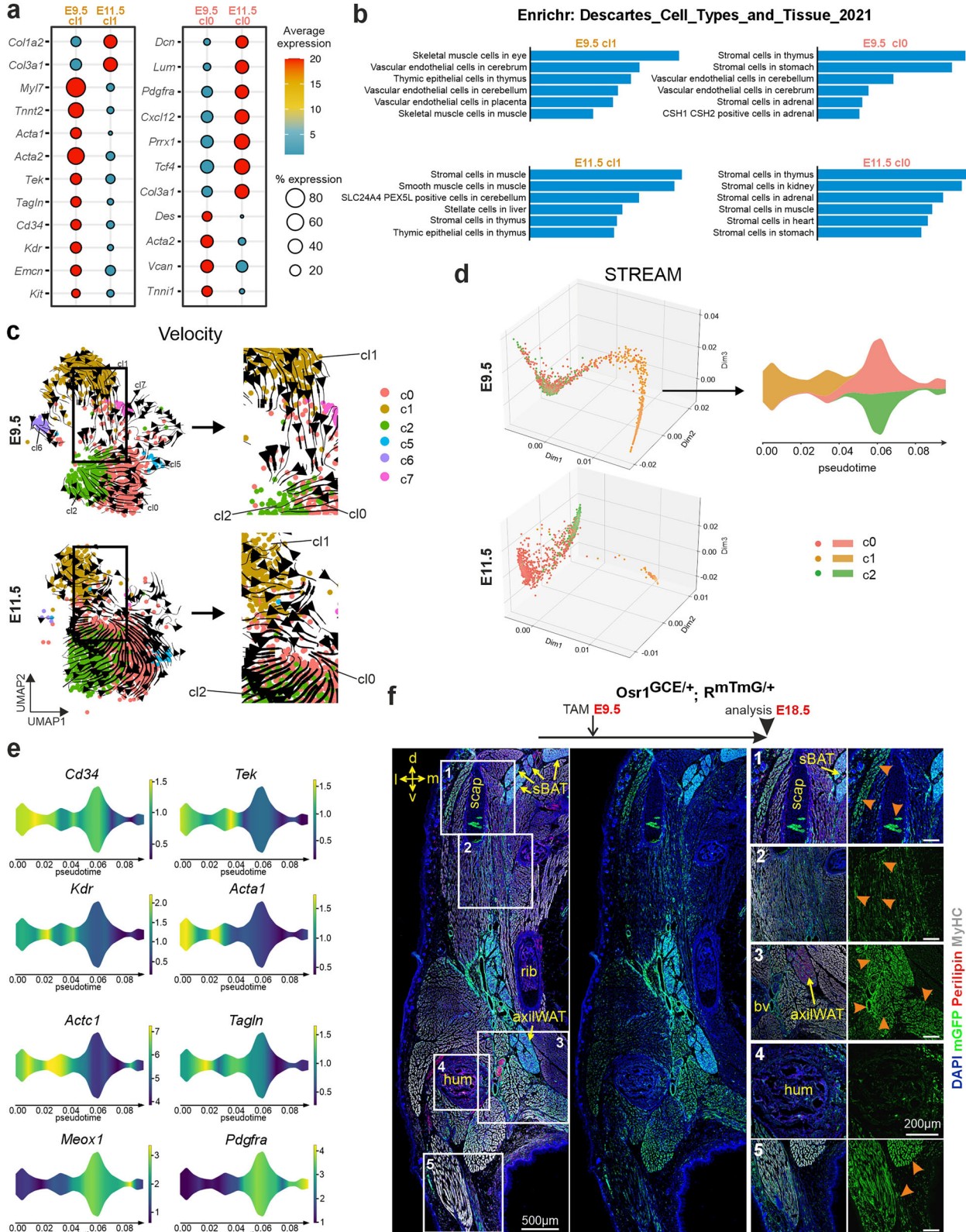

As MABs showed adipogenic potential in vitro[23], this also raises the possibility that the E9.5 source of BAT cells described with our time-restricted lineage tracing experiments (Fig. 3c, d) might correspond to these OSR1+ cells in the DAC, rather than dermomyotome lineage cells with undetectable Osr1-reporter or protein expression at that stage (Fig. 3a and Supplementary Fig. 3a, b). MAB-derived cells were proposed to migrate from the dorsal aorta compartment to their

final sites of tissue integration[23], although this has not yet been demonstrated in vivo. To explore the possibility that Osr1-derived cells may migrate from the anterior DAC to some of the sites where we detected them at E14.5 and E18.5, we used tamoxifen pulse labeling at E9.5 followed by longitudinal analysis at E10.5, E11.5 and E12.5 (Fig. 6c−e). At E9.5, OSR1+ cells locate to the central region of the embryo, mainly surrounding/or labeling the trachea, esophagus, and

**Fig. 5 | Multilineage potential of Osr1⁺ cells at E9.5. a** Dot plot expression analysis of indicated genes in the E9.5 versus E11.5 Osr1-eGFP clusters 1 (left) and 0 (right). **b** Enrichr based gene ontology analysis of Osr1-eGFP clusters 1 (left) and 0 (right). **c** Velocity analysis of E9.5 (top) and E11.5 (bottom) Osr1-eGFP single-cell sequencing datasets without the small clusters 3, 4, 9, 10, 11; boxed region shown as magnification to the right. **d** STREAM inferred trajectories in high-dimensional space visualized for clusters 0, 1 and 2 of the Osr1-GFP E9.5 or E11.5 datasets, respectively. **e** Expression pattern of genes that label specific stages among the trajectory of cl1

into cl0 and cl2 within the Osr1-GFP E9.5 dataset. **f** Genetic lineage tracing analysis of E9.5 Osr1⁺ cells analyzed at E18.5. Tissue section shows the shoulder region; orientation is indicated: d: dorsal; l: lateral; m: medial; v: ventral. Labeling: mGFP (Osr1 lineage; green), Perilipin (adipocytes; red), MyHC (muscle, white) and DAPI (nuclei; blue). Magnifications of boxed regions shown right. Orange Arrowheads indicate E9.5 Osr1 lineage contribution to skeletal muscle, cartilage and blood vessel walls. *AxiWAT* axillary white adipose tissue; *bv* blood vessel; *hum* humerus; *sBAT* subscapular brown adipose tissue; *scap* scapula; representative image of *n* = 3.

the DAC (Fig.3a). Analysis at E10.5 and E11.5 yielded large pools of mGFP⁺ Osr1-descendants on both lateral sides of the dorsal aorta (Fig. 6c, d). In addition, sparse contribution of Osr1⁺ progeny was observed in the aortic endothelium as well, but not in any other endothelial structures (Fig. 6c, d). Intriguingly, individual descendants of E9.5 Osr1⁺ cells appear to have spread laterally from the aortic region, but not from the trachea/esophagus region. These mGFP⁺ cells extended to the most proximal limb buds, closely located to the vicinity of the vasculature (Fig. 6c, d). This supports the idea of migratory behavior of initially DAC-associated Osr1⁺ progenitors to the dorso-lateral periphery between E9.5 and E11.5. Tracing E9.5 Osr1 cells to E12.5, these lateral cells appeared to have further displaced, accumulating in the most lateral flank of the embryo and the proximal limb bud (Fig. 6e). Intriguingly, the connection to the aorta-associated source appeared almost disrupted (Fig. 6e). In addition, at E12.5 strong lineage labeling was found in the trachea/esophagus region (Fig. 6e), likely emanating from expansion of the small pool present at this location earlier (Fig. 6c, d). This altogether suggests that a wave of Osr1⁺ cells leaves the anterior DAC at around E9.5/10.5, overlapping with what has been speculated to be the in vivo behavior of MAB-like cells reviewed in ref. 48.

At E12.5, we also observed first mGFP⁺ cells arriving at the ventromedial portion of the site of presumptive sBAT formation (Fig. 6e). Furthermore, contribution of E9.5 Osr1 descendants to skeletal muscle in the proximal limb could be observed (Fig. 6e), besides contribution to vessel-associated cells and connective tissue cells (Fig. 6e). Both was observed especially in the shoulder / proximal limb region, as we observed before tracing E9.5 Osr1⁺ cells to E18.5 (Fig. 5f).

In contrast, pulse labeling at E11.5 (Fig. 6f) showed a broader distribution of Osr1 descendants in the E12.5 embryo in accordance to the widespread expression of Osr1 at various locations in the embryo at E11.5[24,33,37]. At E11.5, Osr1 is expressed in limb bud mesenchyme already, which is not the case at E9.5 (So and Danielian[37] and Supplementary Fig. 3a). In line, E11.5 Osr1 descendants were traced in E12.5 limbs more distally (Fig. 6f) than E9.5 descendants, which were only found in the proximal limbs (Fig. 6e). Furthermore, E11.5 descendants showed stronger contribution to the sBAT depot already at E12.5 (Fig. 6f) confirming the results at E14.5 (Fig. 3d). In accord with our previous data[24,33] no skeletal muscle fate was observed for E11.5 progeny. Osr1 descendants exclusively contributed to stromal cells as muscle interstitial connective tissue and to vessel-associated cells (Fig. 6f).

Taken together, a subset of E9.5 Osr1⁺ cells expressing MAB markers locates to the dorsal aortic compartment. These cells exhibit migratory behavior between E9.5 and E11.5 and subsequently contribute to diverse tissues in the dorso-lateral part of the mouse embryo, including BAT.

## Discussion

Here, we show that Osr1 is expressed in progenitors for subcutaneous white as well as most brown adipose tissue depots during their emergence in embryogenesis and propose the existence of a previously unrecognized non-dermomyotomal, dorsal aorta-associated early source for brown adipose tissue in vivo.

Brown adipose tissue is thought to predominantly arise from paraxial mesoderm, specifically from the dermomyotomal compartment of the somites[15,16,18,20,31]. In the E11.5 dermomyotome region, OSR1 expression overlaps with the earliest known BAT progenitor marker EBF2[31], and we identify OSR1/PPARγ⁺ presumptive preadipocytes close to the ventromedial edge of the dermomyotome in agreement with a possible displacement of these cells to their target regions. Of note, tracing E11.5 Osr1 descendants showed marked contribution to all BAT depots with the exception of the interscapular depot (iBAT). We show that Osr1⁺ cells are found in the region of the central dermomyotome and ventro-lateral dermomyotome descendants, but not in the dorsomedial dermomyotome and its descendants (Fig. 2). This aligns with a previous report suggesting that the iBAT is formed from dorsomedial rather than central dermomyotomal cells[20], providing a reasonable explanation for the low contribution of Osr1 cells specifically to this depot. Moreover, this emphasizes that iBAT alone is not representative for BAT analysis. This is particularly relevant to humans, as the large iBAT depot present in human infants is not preserved in adults[1]. In parallel, E11.5 Osr1 expression in the limb girdle and limb bud mesenchyme[24] agrees with previously observed origins of subcutaneous white adipose tissue from Prrx1⁺ mesoderm[21].

In contrast, at E9.5, Osr1 is not detectable in the limb bud, the dermomyotomal area, or in dermomyotomal descendants (Fig. 3a and Supplementary Fig. 3a, b), suggesting a distinct origin of Osr1-lineage-derived BAT and subcutaneous WAT cells. In line with previously published mRNA or reporter expression data[36,37,50–52], we found Osr1 expression confined to central tissues as foregut and lung buds in the anterior embryo, in addition we identified the aorta-associated mesenchyme and the aortic endothelium as previously uncharacterized sites of E9.5 Osr1 expression. While our single-cell data show a clear preadipogenic signature in E11.5 Osr1⁺ cells, consistent with OSR1/PPARγ⁺ and OSR1/EBF2⁺ cells in in the dermomyotome area (Fig. 2a, b), this adipogenic signature is diffuse at E9.5. Conversely, at E9.5 a subset of Osr1⁺ cells appear to represent a more multipotent population expressing conflicting marker genes compared to E11.5 cells, which in essence have a stromal character. This parallels observations of Osr1⁺ cells in the developing kidney exhibiting a stepwise restriction of Osr1⁺ cell potential[25]. This also suggests that convergence to a general stromal fibroblastic cell type is a general feature of Osr1⁺ cells at different anatomic locations.

A combination of single-cell sequencing, in situ mapping and lineage tracing suggests that E9.5 Osr1⁺ cells located in the dorsal aortic compartment may comprise a multipotent migratory progenitor correlating to mesoangioblasts (MAB)s.

MABs have attracted significant attention, as MAB-like cells with restricted differentiation potential (smooth and skeletal muscle) can be derived from adult muscle vasculature[53]. In contrast to satellite cells, MABs can cross blood vessel walls, making them attractive therapeutic targets in muscular dystrophies[48]. Of note, MABs have been characterized in vitro, however, these cells have so far not been formally located in the embryo. We found widespread expression of OSR1 in aortic endothelium and the surrounding mesenchyme at E9.5, but only a sparse contribution of the E9.5 Osr1-lineage to aortic endothelium thereafter. This would be in favor of an old hypothesis, suggesting an origin for MABs from the aortic endothelium reviewed in

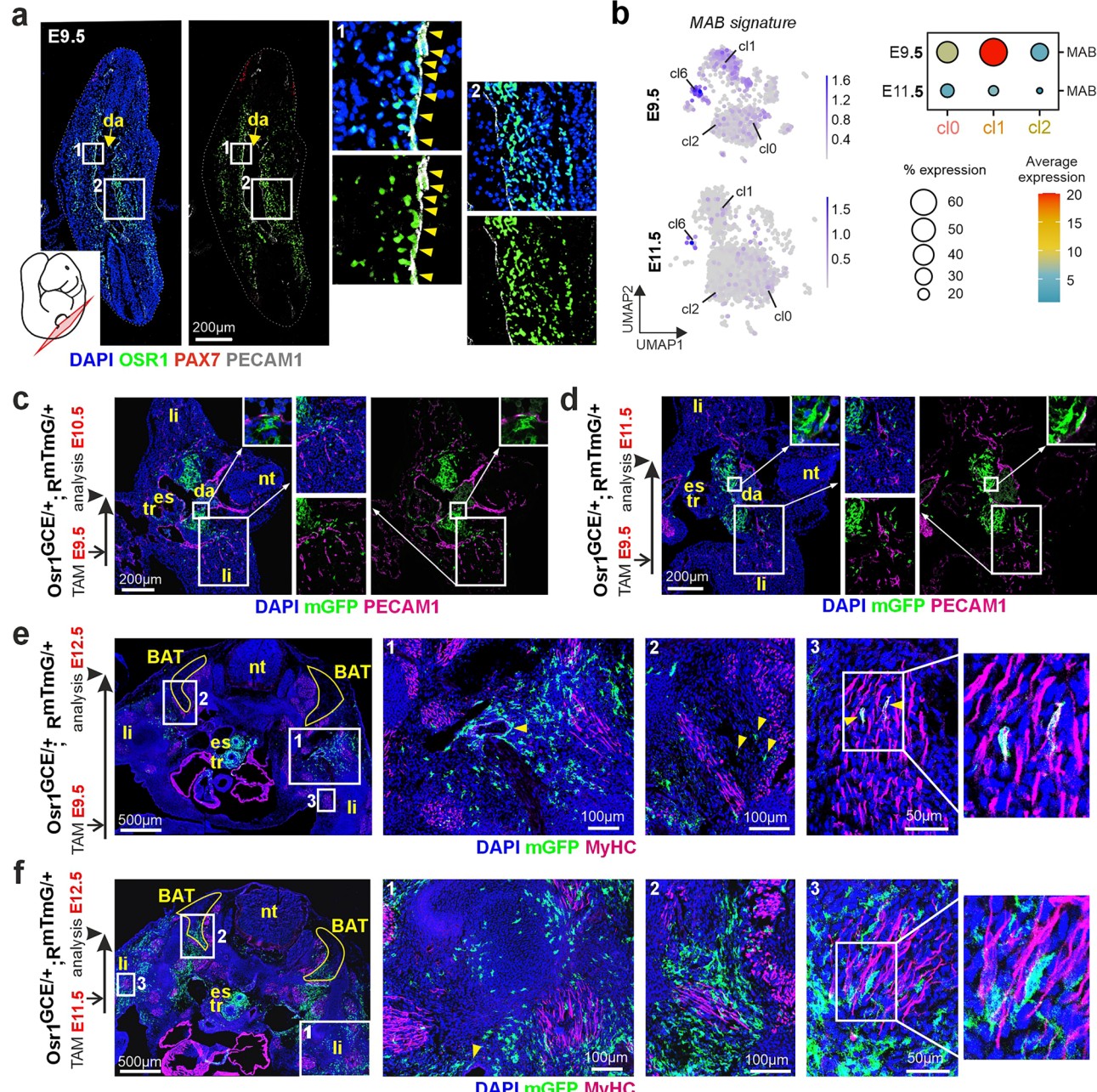

**Fig. 6 | A migratory mesoangioblast-like Osr1⁺ cell population as alternative BAT progenitor at E9.5. a** Tissue sections (section plane indicated) of E9.5 wild-type mouse embryos labeled for OSR1 (green), PAX7 (red), PECAM1 (endothelial cells, white) and DAPI (nuclei; blue). Boxed areas are shown as magnifications right, arrowheads indicate PECAM1⁺/OSR1⁺ cells; representative image of *n* = 3. **b** Feature plot (left) and dot plot (right) depiction of mesoangioblast (MAB) signature mapped to the Osr1 datasets. **c–f** Tissue sections of Osr1^GCE/+;R^mTmG/+ embryos induced with tamoxifen at E9.5 (C-E) or E11.5 (F) and analyzed at indicated time points (see schematic depictions to the left). **c**, **d** are labeled for mGFP (Osr1 lineage; green), PECAM1 (magenta) and DAPI (nuclei; blue), (**e**, **f**) are labeled for mGFP (Osr1 lineage; green), MyHC (skeletal muscle; magenta) and DAPI (nuclei; blue). BAT: presumptive area of brown adipose tissue; da: dorsal aorta; es: esophagus; li: limb bud; nt: neural tube; tr: trachea. Arrowheads indicate Osr1 lineage cells associated with blood vessels in e1 and f1, Osr1 lineage cells within the presumptive BAT depot in e2, and Osr1 lineage contributing to myofibers in e3. Representative images of *n* = 3 for each condition.

ref. 48. Support for this hypothesis came from a study showing the contribution of Flk1(Kdr)⁺ progeny to skeletal muscle[54]. In line with this, we observe *Kdr* expression in E9.5 cl1 Osr1⁺ cells in our single-cell dataset.

Time-resolved Osr1 lineage tracing experiments indicate that at least a fraction of aorta-associated E9.5 Osr1⁺ cells leaves this niche at around E10.5 to migrate dorso-laterally, reaching the nascent sBAT depot at E12.5. Intriguingly, the E11.5 Osr1⁺ pool contributes earlier, and in increased fashion, to this depot (Figs. 3d, 6e, f). A simple explanation

would be that E9.5 cells arising at the dorsal aorta need to be further displaced to reach this target region compared to cells from the nearby dermomyotome region. These findings support our hypothesis that the BAT contribution of E9.5 Osr1⁺ cells does not account to sub-detection levels of Osr1^GCE expression in the dermomyotome or its derivates, but that tamoxifen labeling indeed targets two separate populations at E9.5 versus E11.5.

Furthermore, E9.5 Osr1⁺ cells appear multipotent in vivo, as their descendants contribute to a variety of mesodermal tissues as adipose

tissue, cartilage, smooth muscle or skeletal muscle in the anterior trunk and proximal limbs in line with MAB in vitro potential[23,46]. We note that while our in vivo lineage-tracing data provide strong evidence for the multilineage differentiation potential of E9.5 Osr1+ cells, complementary in vitro validation would further strengthen these findings. Although we attempted to culture E9.5 Osr1+ cells according to protocols used for MABs[23,46], we were unable to successfully propagate them under the conditions tested. Further adaptations to perform cell culture differentiation experiments, such as a three-dimensional context, might be required[55].

In summary, our findings align with and provide in vivo evidence for previous assumptions about MAB-like cells in vivo. It was proposed that MAB-like cells might be a transient population of vascular-derived progenitors existing around E9.5/10.5 in the mouse, which then could possibly migrate from the anterior DAC to their target sites in the following 2-3 days[23], a time window and behavior aligning with E9.5 Osr1+ cells and their descendants.

This stratifies the origins of BAT and subcutaneous WAT, and agrees with the assumption that BAT and WAT depots at different anatomical locations recruit variable and location-specific progenitor pools for their developmental establishment[6,12,18,20]. Long-term lineage tracing of E11.5/12.5 Osr1+ cells indicated contribution to the adipose tissue perivascular niche. This, together with the persistent contribution of developmental Osr1 cells to adult depots, suggests that embryonic Osr1+ cells comprise not only developmental pre-adipogenic cells participating in depot initiation, but also progenitors for adult adipose tissue-resident stem/progenitor cells (ASPCs). Unfortunately, the relative contribution of the E9.5 and E11.5 pools to this population cannot be disentangled as single-dose TAM induction proved ineffective for long-term labeling in the Osr1GCE model in our hands[33]. However, given that both E9.5 and E11.5 Osr1+ cells contribute to perivascular tissue early on (Fig. 6e, f), contribution from both sources appears possible.

The expression of Osr1 in developmental adipogenic progenitors of different types at different locations suggests a common role for Osr1 in brown and white adipogenic differentiation rather than specifying a distinct subtype. Moreover, the progenitor populations in adult subcutaneous WAT have been further stratified, leading to the discovery of a specific population of adipogenic-regulatory cells (AREGS) characterized by expression of *F3* encoding CD142[56]. A similar population has been identified in intramuscular adipose tissue[57]. AREGS suppress adipogenesis in a paracrine manner and intriguingly show high expression of *Osr1*[56,57]. *F3* expression was not detectable in our E9.5 dataset, at E11.5 *F3* was detected in cl0/2 in a dispersed pattern not matching the pre-BAT module (Supplementary Fig. 9a). It remains unknown, however, if *F3*/CD142+ cells function as AREGS in the embryo. Comparing *Osr1* and *F3* expression in the aortic BAT datasets[27] confirmed partial overlap, yet *Osr1* showed a broader expression suggesting that Osr1 is not an exclusive AREG marker but plays a broader role (Supplementary Fig. 9b). We recently found that loss of Osr1 compromises the adipogenic differentiation of adult mouse muscle interstitial fibro-adipogenic progenitors suggesting its expression is necessary for adipogenesis[58]; it will be interesting to test whether Osr1 fulfills an overlapping function in development.

## Methods

### Animal experimentation and ethical approval
Animal experiments were performed in accordance with the European Union legislation for the protection of animals used for scientific purposes and approved by the Landesamt für Gesundheit und Soziales Berlin under license numbers ZH120, G0240/11, G0268/16 and G0250/17. Mouse lines were maintained in an enclosed, pathogen-free facility at a temperature of 22 °C, 55% of humidity, and 12 h light / dark cycles. All Animals were euthanized using a GasDocUnit® (Medres Medical Research) followed by cervical dislocation. Female mice aged

8–24 weeks were bred for experiments involving embryos; embryonic stages used are indicated in all figure panels. 11-week-old mice were used for all experiments involving adult subjects. Allocation to experimental groups was performed according to genotype; sex-based analysis was not performed due to low animal numbers. The mouse lines used in the study have been previously described: Osr1GCE[25], R26RmTmG[30], Osr1LacZ[38], Myf5Cre[59], Pax7CreERt2[60], mice were maintained on a C57Bl6 background.

### Tissue preparation
Embryonic tissues underwent fixation in 4% paraformaldehyde (PFA) for 2 h on ice for developmental stages ranging from E12.5 to E18.5, and for 1 h for earlier embryonic stages. Following fixation, tissues underwent dehydration in 15% and 30% (w/v) sucrose solutions (Roth). Subsequently, cryo-embedding in O.C.T. compound (Sakura) was carried out in a chilled ethanol/dry ice bath. Sectioning of embryonic tissue was performed at thicknesses of 12 micrometers. Adult adipose tissues were fixed in 4% formalin overnight at room temperature. Tissues were dehydrated through a series of increasing ethanol concentrations and Ultraclear (J.T.Baker) to paraffin at 60 °C. Sectioning of paraffin-embedded tissue was performed at a thickness of 6 micrometers.

### Immunolabeling
Paraffine sections: To prepare slides, paraffin removal and tissue rehydration were achieved with a 1 h treatment using UltraClear. This was followed by an ethanol series (100%, 95%, 70%, 30%) and a wash with distilled water. Tissue sections were then treated for 15 minutes with blocking buffer (4.11 g/l Citric acid, 10.76 g/l $Na_2HPO_4 \cdot 2H_2O$, add bidest $H_2O$ to 1 l) containing 1,5% $H_2O_2$ to inhibit endogenous peroxidases, followed by rinsing in distilled water. Antigen epitopes were unmasked using blocking buffer II (242 g/l Tris-Base, 18.6 g/l EDTA, pH 9 (with HCl), add bidest $H_2O$ to 1 l. Dilute for use 1:50 in $H_2O$) and microwave heating, and autofluorescence was reduced with 0.3 % Sudan Black in 70 % EtOH. After a 1 h blocking step with 3% BSA/PBS, primary antibodies in 1%BSA/PBS were applied overnight at 4 °C. Slides were washed with PBS, and secondary antibodies, along with DAPI, were applied for 1 h at room temperature. After washing with PBX (100 ml of 10x PBS solution. 1 ml of TritonX-100 ad 1 l distilled water), and sections were mounted with Fluoromount-G (Biozol) and stored at 4 °C in the dark. Frozen sections: Cryo-sections were thawed: 30 minutes at room temperature in an unopened box, followed by 15 min on a 37 °C heating board. A 10 min wash in 1xPBS removed the O.C.T. matrix. Subsequent steps were conducted in a humid chamber. Sections were treated with 0.4% Triton X-100 for 10 min at room temperature for permeabilization, then blocked with 3% BSA/PBS for 1 h. Primary antibodies, diluted in 1% BSA/PBS, were applied overnight at 4 °C. Primary antibody dilutions used: PPARγ: 1:100; OSR1: 1:100; GFP: 1:800; FABP4: 1:300; PAX7: 1:100; MyHC: 1:800; EBF2: 1:20; PECAM1: 1:250; alpha-SMA: 1:250; Perilipin A/B: 1:300; UCP1: 1:200. After three 10-minute washes in 1xPBX, secondary antibodies (1:500) and DAPI (1:1000) were applied for 1 h at room temperature. Three 10 min washes in 1xPBX removed excess antibodies. Slides were mounted with Fluoromount-G and stored at 4 °C in the dark. All antibodies are listed in Supplementary Table 1.

### LacZ staining
After removing the O.C.T. from the cryo-sections using 1xPBS, an additional PBX washing step was performed to permeabilize the tissue sections. Subsequently, the slides were immersed in a staining solution (0,01% sodium deoxycholate, 0.02% NonidetTM P40, 0.4 mg/ml MgCl2, 1.7 mg/ml K3Fe(CN)6, 2.1 mg/ml K4Fe(CN)6 • 3H2O, 1 mg/ml X-Gal-substrate). Incubation was conducted in the dark at 37 °C, with the staining process lasting between 3 to 8 h. The following day, slides were washed with 1xPBS, mounted with Fluoromount-G, and stored in the dark at 4 °C.

### Oil red O staining

O.C.T. was removed from tissue sections using 1xPBS. A 0.5% ORO stock solution in isopropanol was prepared and diluted to 0.3% with distilled water. After resting for 10 min and filtering, slides were rinsed in 60% isopropanol and stained for 30 min in ORO solution at room temperature. Excess staining was removed with a short rinse in 60% isopropanol. Finally, tissues were rinsed with 1xPBS, mounted with Fluoromount-G, and stored at 4 °C.

### Imaging

Confocal images of immunolabeled sections were acquired utilizing Zeiss laser scanning microscope systems LSM710, LSM810 and Cell-discoverer 7, or the Leica DMi8 epifluorescence microscope equipped with optical deconvolution. Images were captured using ZEN (Zeiss) or LAS-X (Leica) software.

### Genetic lineage tracing

CreERt2 recombinase activation was achieved by tamoxifen induction at specific timepoints. A dosage of 3 mg tamoxifen / 40 g body weight was intraperitoneally injected into the abdominal space of pregnant mice. To alleviate tamoxifen side effects on pregnancy, 2 mg progesterone / 40 g body weight was applied when traced from E9.5 to E18.5 or when traced with a tamoxifen (1.2 mg / 40 g body weight) injection on two consecutive embryonic stages E11.5 + E12.5 to adulthood (11weeks).

### Icell8 single-cell RNA-seq analysis (Takara)

For single-cell mRNA-Seq analysis, GFP+ cells from 14 E9.5 and 6 E11.5 embryos of the Osr1[GCE/+] line were isolated and pooled according to stage. The isolation of GFP+ cells was performed according to a published protocol[33]. Embryos were individually dissociated in 2 ml tubes containing 400 µl 1 ml high-glucose Dulbecco's modified eagle medium (DMEM, Pan Biotech) containing 10% fetal bovine serum (FBS, Pan Biotech) and 1% penicillin/streptomycin (P/S) solution and 3 µl of Collagenase A (100 µg/µl). The embryos were minced and incubated in a pre-warmed thermomixer at 37 °C for 5 min for E9.5 embryos and 10–15 min for E11.5 embryos. Following digestion, the cell suspension was gently pipetted up and down and then centrifuged at 300 x g for 5 min. The supernatant was discarded, and the cell pellet was washed with 1 x PBS. After another centrifugation step, the 1 x PBS was replaced with FACS sorting medium containing 1% FCS, 2 mM EDTA mixed in 1 x PBS (filtered). Before FACS, the cell suspensions were strained using a 40 µm mesh to enable cell separation. The resulting cell suspension was further purified using a FACS Aria III (BD Biosciences), sorting Osr1-GFP+ cells into a landing buffer consisting of growth medium (DMEM High Glucose supplemented with 10% FBS, L-Glutamine, Pen/Strep). The gating strategy is depicted in Supplementary Fig. 4. The sorted cells were centrifuged at 300 × g for 5 min at room temperature, and the pellet was resuspended in 1% BSA in PBS (2 µ/µl−1). The single-cell suspension was processed using the SMARTer iCell8 Single Cell System (Takara). First, the cell suspension was stained with a 1:1 mixture of Hoechst 33342 and Propidium Iodide according to the manufacturer's instructions. The stained cells were then loaded into a 384-well source plate (Takara) and dispensed into the iCell8 SmartChip (Takara) using the iCell8 MultiSample NanoDispenser (iCell8 MSND, Takara; 1st dispense), following the manufacturer's protocol. The nano-wells were imaged using the iCell8 Imaging System (Takara), and images of all 5,184 nano-wells were acquired. Using the CellSelect software, nano-wells containing high-quality single cells were selected. From the E9.5 pool, 1282 nano-wells containing cells were chosen, and 1609 nano-wells were selected from the E11.5 pool. After cell selection, the full-length SMART-Seq protocol was followed. The MSND dispensed 35 nl of RT-PCR master-mix (iCell8 SMART-Seq - Reagent kit, Takara; 2nd dispense) into each nano-well. Following the RT reaction, 35 nL of forward indexing primers (SMART-Seq iCell8 -

Indexing Primers, P5; 3rd dispense) were added. The tagmentation protocol (4th dispense) was then applied with the addition of 35 nL of tagmentation mix (iCell8 SMART-Seq Reagent kit, Takara) to each nano-well. Afterward, 35 nL of reverse indexing primers (SMART-Seq iCell8 – Indexing Primers, P7, 5th dispense) were added. The SmartChip was then placed in the iCell8 Chip Cycler, and Library PCR 1 was performed according to the manufacturer's instructions. After the first PCR, libraries were extracted from the chip using the collection module (iCell8 SMART-Seq – Chip kit) and purified twice using AMPure XP beads (1:1 ratio, Beckman Colter). A second PCR (Library PCR 2), followed by another round of AMPure XP purification (1:1), was performed to generate the final full-length, sequencing-ready libraries. Libraries were validated and quantified using Qubit (dsDNA HS Assay Kit, Thermo Fisher Scientific), ScreenTape (D5000, Agilent) and KAPA Library Quantification (KAPA Biosystems, Roche). The final libraries were sequenced on a HiSeq-4000 with dual indexing and a 2 × 75 paired-end configuration.

### Icell8 Data pre-processing

Raw sequencing data was demultiplexed using bcl2fastq v2.20.0. The resulting reads from the FASTQ files were then aligned to the mouse reference genome (GRCm38) using STAR v2.7.3a[61] according to the STARsolo workflow, with the following settings: --soloFeatures Gene GeneFull SJ Velocyto, -- soloAdapterMismatchesNmax 2, and --soloCBmatchWLtype 1MM.

### Seurat processing

The E9.5 and E11.5 datasets were analyzed using Seurat v3[62]. First, Seurat objects were created with CreateSeuratObject (min.features = 200, min.cells = 3). The data was log-normalized using NormalizeData, followed by identification of 2000 variable features with FindVariableFeatures (selection.method = "vst").

Data integration was performed using the FindIntegrationAnchors and IntegrateData functions (dims = 1:30). After this step, the integrated object was scaled with ScaleData, and cell cycle scores were regressed (vars.to.regress = c("S.Score", "G2M.Score")).

Principal component analysis (PCA) was carried out with RunPCA using 30 components, which were subsequently used to generate a Uniform Manifold Approximation and Projection (UMAP) reduction with the RunUMAP (dims = 1:30) function. Ultimately, the neighbor graph was constructed with FindNeighbors (dims = 1:30), followed by clustering using the Louvain algorithm with FindClusters (resolution = 0.5).

### Module scoring workflow

Module scores were computed using the AddModuleScore function in Seurat v4.1.0. Gene sets were derived from[43] by selecting the top 50 significantly upregulated genes (p_val_adj < 0.05), ranked by average log2 fold change (avg_log2FC), from the following clusters: "Tendon progenitors", "Irregular connective tissue", "Muscle" (referred to as "Myogenic" in our analysis), "Mesenchyme", "Early digit condensation", and "Proximal condensation". An additional "Cartilage condensation" gene set was generated by taking the union of the "Early digit condensation" and "Proximal condensation" sets, resulting in a total of 100 differentially expressed genes.

### Downstream analysis using Stream and velocity

Velocity analysis: Loom files were generated for each timepoints from BAM files that were aligned and sorted (velocyto_run, v0.17.17)[63], and the subsequent files were merged (loompy.combine). From here, count matrices from each timepoints were processed separately with scVelo (v0.2.3)[64]. Filtering and normalization were performed using on the top 5000 highly variable genes with scvelo.pp.filter_and_normalize (min_shared_counts = 30, n_top_genes = 5000). Moments were computed for each cell across their nearest neighbors within the PCA space

(default parameters within scvelo.pp.neighbors, scvelo.pp.moments). Ultimately, velocities specific to each gene were determined through the stochastic model (scv.tl.velocity), and the velocity graph was built (scvelo.tl.velocity_graph). Trajectory inference using Stream: Trajectory inference was performed independently for E9.5 and E11.5 on clusters c0, c1 and c2 using STREAM (v1.1)[65]. Three major steps are performed within the STREAM pipeline: variable features selection (default parameters within select_variable_genes), dimensionality reduction (dimension_reduction with n_components = 3, n_neighbors = 15), and simultaneous tree structure learning and fitting using Elastic Principal Graph (ElPiGraph) algorithm (seed_elastic_principal_graph with n_clusters = 10, followed-up by elastic_principal_graph with epg_alpha = 0.03, epg_mu = 0.05, epg_lambda = 0.05). For the E9.5 timepoint, transition markers were identified by computing Spearman's rank correlation between pseudotime and gene expression along the single S0-S1 branch (detect_transition_markers with cutoff_spearman = 0.1, cutoff_logfc = 0.25). This yielded 7834 genes that have variable gene expression across pseudotime.

## Statistical analysis and reproducibility

In Figs. 1e and 3c, three Osr1$^{GCE/+}$;R$^{mTmG/+}$ E18.5 embryos were investigated for mGFP contribution to sBAT, cBAT and iBAT. Depending on the size of the BAT depot, one to eight cross sections per depot of each embryo were quantified. The total number of DAPI$^+$/FABP4$^+$ adipocytes and the subset of DAPI$^+$/FABP4$^+$/mGFP$^+$ cells in percent was assessed. No statistical method was used to predetermine sample size. No data were excluded from the analysis. The experiments were not randomized; the investigators were not blinded to allocation during experiments and outcome assessment.

## Reporting summary

Further information on research design is available in the Nature Portfolio Reporting Summary linked to this article.

## Data availability

Single-cell RNA-seq data generated in this study are available via the BioProject accession number PRJNA1186658. The processed single-cell object is available via Zenodo [https://doi.org/10.5281/zenodo.14170959]. This paper analyzes existing, publicly available data: GSE119945[40] [https://www.ncbi.nlm.nih.gov/geo/query/acc.cgi?acc=GSE119945] and GSE164528[27] [https://www.ncbi.nlm.nih.gov/geo/query/acc.cgi?acc=GSE164528]. All other data generated in this study are provided in the Supplementary Information and Source Data file. Source data are provided in this paper.

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

## Acknowledgements

This work was funded by a grant from the Deutsche Forschungsgemeinschaft (DFG) to S. Stricker (STR 854/7-1). T.J.S. was supported by a grant within the German Center for Diabetes Research (DZD), funded by the German Federal Ministry of Education and Research (BMBF) and the State of Brandenburg (DZD grant IDs 82DZD03E6G, 82DZD03C3G, and 82DZD03D03, to T.J.S). We are gratefully to Andrew P. McMahon (Keck School of Medicine of USC, USA) and Andreas Kispert (Hannover Medical School, Germany) providing Osr1$^{GCE}$ and R$^{mTmG}$ mice. We thank Françoise Helmbacher for critically reading the manuscript. We thank C. Bräuning (FACS core facility of the MDC in the Berlin Institute for Medical System Biology (BIMSB)) for isolating Osr1-eGFP cells for single-cell analysis.

## Author contributions

Conceptualization: S. Stricker. Methodology: S. Stricker, S. Sauer, T.J.S., T.C., and C.G.T. Software: C.F., A.K.S., T.C., and S. Sauer. Formal Analysis: S.H., C.F., and A.K.S. Investigation: S.H. performed the majority of experiments and data collection. P.V.G., G.K., Z.G.M., and V.P. performed additional data collection. C.F., A.K.S., and S.H. performed single-cell analysis. Resources: C.G.T., T.C., T.J.S., S. Sauer, and S. Stricker. Data Curation: C.F., A.K.S., T.C., S. Sauer, and S. Stricker. Writing – Original Draft: S.H. and S. Stricker. Writing – Review & Editing: S.H., S. Stricker, and T.J.S. Visualization: S.H., C.F., A.K.S., and S. Stricker. Supervision: S. Stricker, T.J.S., T.C., and S. Sauer. Project administration: S. Stricker, S. Sauer, T.C., and T.J.S. Funding Acquisition: S. Stricker, T.C., and T.J.S.

## Funding

## Competing interests

The authors declare no competing interests.
