## [Transparent Peer Review file · Nature Communications]

The dorsal aortic compartment is a developmental source of brown adipose tissue in mice

Corresponding Author: Dr Sigmar Stricker

Version 0:

Reviewer comments:

Reviewer #1

(Remarks to the Author)

This manuscript presents a well-organized and important study of BAT development, particularly focusing on Osr1-expressing cells. The work provides new insights into the distinct developmental origins of different BAT depots and multipotent mesenchymal progenitors expressing Osr1 at E9.5. While the overall experimental design and data presentation are commendable, the multi-lineage potential of Osr1+ cells at E9.5 would benefit from additional work as outlined below.

1. Lines 134-137: Please include quantification data for sBAT, cBAT, and iBAT.
2. Line 251 (Fig.3): The similarity in single-cell data between E9.5 and E11.5 is interesting and warrants further discussion. Given that cl0-2 in E9.5 are described as multi-lineage cells, it would be valuable to provide evidence of non-adipogenic cells appearing in the E11.5 data.
3. Line 427: The manuscript could benefit from incorporating MOSTA atlas data for Osr1 at E9.5, E11.5 and E12.5 to visualize the various localization of Osr1+ cells.
4. To build on the current findings, additional validation experiments showing how E9.5 Osr1+ cells contribute to adipogenic and non-adipogenic cells would be valuable. The experimental approach using only Osr1-EGFP positive cells is in my opinion not sufficient. The authors could consider using Osr1GCE/+ mice crossed with Rosa26mTmG mice for single-cell analysis or FACS to demonstrate the multi-lineage potential of Osr1+ cells.
5. It would be interesting to explore whether the distinct developmental origin of iBAT compared to cBAT and sBAT correlates with different functional roles or regulatory mechanisms. Would it be possible to analyze the existing data to predict certain features?

Reviewer #2

(Remarks to the Author)

This manuscript reports an alternative source of progenitor cells that give rise to brown adipocytes, the Osr1+ cells that are multipotent and appear at E9.5. This population is located in the dorsal aortic compartment and is proposed to be an early source of brown adipocytes during embryonic development. Interestingly, scRNA-seq data indicates that this population may correlate with MABs.

The discovery of two distinct Osr1+ cell populations at E9.5 and E11.5 is quite interesting and potentially shows a new lineage of brown adipocytes that is time- and location-specific. I have a few comments that will hopefully improve the manuscript.

1. It would be nice to isolate Osr1+ cells at E9.5 and E11.5 to validate that these cells from E9.5 are multipotent, using differentiation protocols for brown adipocyte, white adipocyte, cartilage, and potentially muscle cells.
2. Do Osr1+ cells at E9.5 reduce Osr1 expression at E11.5?
3. Perilipin staining could be added as a more specific marker for adipocytes, at least in the major images showing BAT, as FABP4 is expressed in many other cell types. Also, UCP1 could be used to label brown adipocytes.
4. Please include more discussion about the differences observed in different BAT depots. Do they potentially have different functions in thermogenesis/metabolism?

Reviewer #3

(Remarks to the Author)

The manuscript provides a compelling and innovative perspective on BAT developmental biology by identifying the dorsal aortic compartment as a novel source of BAT progenitors. The use of advanced tools, including single-cell RNA sequencing, lineage tracing, and immunohistochemistry, adds rigor and credibility to the findings. The hypothesis that E9.5 Osr1+ progenitors are distinct from dermomyotomal progenitors is clearly articulated and well-supported.

However, additional functional validations would strengthen the manuscript and further support the conclusions. The study convincingly describes the migratory behavior and multipotency of E9.5 Osr1+ progenitors, yet the functional evidence for their differentiation potential remains limited as well their implication in controlling adipogenesis. To further enhance the manuscript, I recommend the following:

1) Validate Multipotency of E9.5 Osr1+ Progenitors:

Perform in vitro differentiation assays to test the capacity of these progenitors to differentiate into adipogenic, myogenic, and other mesodermal lineages. Alternatively, conduct transplantation experiments to evaluate their contribution to BAT and other lineages in vivo.

2) Assess Thermogenic Potential of BAT derived from Osr1+ progenitors:

Evaluate the thermogenic activity of BAT derived from E9.5 Osr1+ progenitors using molecular markers like UCP1 or through metabolic assays could address this point.

3) Clarify the Role of Osr1 in Adipogenesis:

Use Osr1 knockout or knockdown models to investigate its specific role in regulating adipogenesis. Examine whether Osr1+ progenitors include subpopulations that overlap with CD142+ adipogenesis-regulatory cells (Areg) previously identified in adult adipose depots or mesoangioblasts (Schwalie et al., Nature 2018, 559(7712):103-108; Camps et al., Cell Rep. 2020, 31(5):107597)

These additional experiments will substantiate the authors' conclusions and further elucidate the role of Osr1+ progenitors in BAT development and adipogenesis.

Version 1:

Reviewer comments:

Reviewer #1

(Remarks to the Author)

None

Reviewer #2

(Remarks to the Author)

The authors had addressed all the concerns, and there are no further comments.

Reviewer #3

(Remarks to the Author)

The authors have substantially revised the manuscript and have responded carefully and constructively to the points raised in the first review. The additional analyses, newly included immunostainings, integration of MOSTA atlas data, and expanded discussion significantly improve the clarity and robustness of the study. Overall, the manuscript is now much stronger and more balanced.

- They have clearly explained the technical limitations related to tamoxifen-induced lethality and low recombination efficiency. The decision to move the E18.5 lineage-tracing data to the Supplementary Information and to temper quantitative language is appropriate. The revised text now reflects the limitations transparently and adequately.

- While direct in vitro differentiation of E9.5 Osr1+ cells is still missing, the authors provide a reasonable explanation of the technical limitations and include a clear statement acknowledging this in the Discussion. Given the constraints, the extensive in vivo lineage tracing is acceptable evidence. The response is satisfactory for this manuscript, with the understanding that mechanistic exploration may follow in future work.

REPLIES TO REVIEWER COMMENTS

Reviewer #1 (Remarks to the Author):

This manuscript presents a well-organized and important study of BAT development, particularly focusing on *Osr1*-expressing cells. The work provides new insights into the distinct developmental origins of different BAT depots and multipotent mesenchymal progenitors expressing *Osr1* at E9.5. While the overall experimental design and data presentation are commendable, the multi-lineage potential of *Osr1*⁺ cells at E9.5 would benefit from additional work as outlined below.

1. Lines 134-137: Please include quantification data for sBAT, cBAT, and iBAT.

We apologize, but due to the very low survival rate of Tamoxifen-injected litters in our hands, we were only able to obtain two biological replicates despite considerable effort. Therefore, a quantitative analysis appears not meaningful in this case. We have clearly stated this limitation in the revised manuscript (p.5, lines 139-142; “Due to high mortality of tamoxifen-pulsed fetuses after birth, we were only able to retrieve 2 biological replicates, thus data were not quantified. This nevertheless indicated persistent contribution of *Osr1* descendants to sBAT, cBAT and ingWAT, but also to adult aortic BAT (aBAT which is not detectable in the embryo) as well as to adult iBAT (Supplementary Fig. 2a).”) and have revised the text to interpret these findings more cautiously, removing all quantitative statements. Given these constraints, we have moved the corresponding data to the Supplementary Information. Nevertheless, we believe that these results remain valuable, as they show long-term contribution of *Osr1*-embryonic cells to adult adipose depots and their vascular niche.

We would also like to emphasize that quantification of lineage-tracing data obtained using a CreERT2 system is inherently limited, as it depends on tamoxifen dosage, administration duration, and the efficiency of the CreERT2 allele. As discussed in the manuscript, our labeling strategy using the low-efficiency *Osr1*-CreERT2 allele was deliberately designed to maximize specificity rather than labeling efficacy. Consequently, any quantitative interpretation of these data must be made with caution.

2. Line 251 (Fig.3): The similarity in single-cell data between E9.5 and E11.5 is interesting and warrants further discussion. Given that cI0-2 in E9.5 are described as multi-lineage cells, it would be valuable to provide evidence of non-adipogenic cells appearing in the E11.5 data.

Thank you for this comment and suggestion. We in fact mainly describe the E9.5 cI1 population as multipotent, as it expresses a combination of stromal, skeletal/cardiac muscle, smooth muscle, and angiogenic markers, which are largely absent from the E11.5 dataset (now Fig. 5a, S7b). Velocity and STREAM analyses indicate that these cells partially contribute to the E11.5 cI0/2 populations, which predominantly acquire a stromal progenitor identity. To emphasize this point, we have moved former Supplementary Figures 4a, b to the main figures (now Fig. 5a, b).

To further analyze this, we used module scores corresponding to myogenic, tendon, and cartilage condensation clusters from Rouco et al. (2021, Nat. Commun. 12: 7140; doi:10.1038/s41467-021-27492-1) and mapped these to our dataset. This analysis confirmed that a myogenic signature is confined to E9.5 cI1 and subsequently lost, while tendon and cartilage signatures are found in E9.5 cI1, and persist at E11.5 in cI0/2 (new Supplementary Fig. 6c). Consistently, key tendon and cartilage transcription factors (*Scx*, *Sox9*, *Runx2*) are expressed in the E11.5 cI0/2 (new Supplementary Fig. 6d).

These findings align well with our previous observations. We have previously shown that E13.5 *Osr1*⁺ cells still express low levels of *Scx*, *Sox9*, and *Runx2*, and that *Osr1* itself represses alternative differentiation programs (Vallecillo-García et al., 2017, Nat. Commun. 8: 1130; doi:10.1038/s41467-017-01120-3). In *Osr1* knockout cells, these transcription factors are upregulated, and *Osr1* KO cells display chondrogenic potential, consistent with prior findings that *Osr1* must be downregulated to permit chondrogenic differentiation (Liu et al., 2013, PNAS 110: 14452; doi:10.1073/pnas.1306495110). Together, these data suggest that cells derived from E9.5 *Osr1*⁺ progenitors that have undergone overt differentiation into alternative lineages have downregulated *Osr1* expression by E11.5 and thus are not represented in our dataset representing *Osr1*-GFP⁺ expressing cells.

In summary, while the broad multipotent signature observed in E9.5 c1 is no longer detectable among *Osr1*-GFP⁺ cells at E11.5, we fully agree with the reviewer that the E11.5 c10/2 populations retain progenitor characteristics and display a more restricted set of lineage-specific markers.

This is discussed in the manuscript on p.15 lines 364-377 (“To further investigate the lineage identities in our *Osr1* dataset, we used embryonic mouse limb single cell sequencing data⁴³ and derived myogenic, tenogenic and chondrogenic module score signatures. While the myogenic signature exclusively mapped to E9.5 c1 in line with the analysis above, tenogenic and chondrogenic signatures mapped predominantly to c1 at E9.5 and shifted to c10/2 at E11.5 (Supplementary Fig. 6c) in line with a hypothesized flow of cells in this direction. Expression of key marker genes that describe a tenogenic (*Scx*) as well as osteo-chondrogenic (*Sox9* and *Runx2*) identity were detected in an overlapping pattern (Supplementary Fig. 6d). Intriguingly, we have shown before that E13.5 *Osr1*⁺ cells still have expression of alternative lineage genes as *Scx* and *Runx2*, but that chondrogenic, tenogenic and also myogenic differentiation is actively repressed by *Osr1* itself^{24,44}; indeed, *Osr1* is a direct repressor of *Sox9*⁴⁵. Comparing *Osr1* expression levels between the E9.5 and E11.5 datasets suggested an upregulation of *Osr1* over time as cells progress to an overall stromal fate (Supplementary Fig. 6e). This would agree with the assumption that a lower level of *Osr1* expression in *Osr1*⁺ cells at E9.5 may be instrumental to maintain a multipotent fate, which becomes more and more restricted as *Osr1* expression increases.”).

3. Line 427: The manuscript could benefit from incorporating MOSTA atlas data for *Osr1* at E9.5, E11.5 and E12.5 to visualize the various localization of *Osr1*⁺ cells.

Thank you very much for this helpful suggestion. MOSTA data for E9.5 (Fig. 3b) and E11.5 (Fig. 4e) were already included in the manuscript. We have revisited additional sections of the MOSTA E9.5 and E11.5 datasets; however, these did not yield further relevant insights, as the MOSTA atlas primarily provides central sagittal sections and therefore does not include structures such as the limbs.

In response to the reviewer’s suggestion, we have now incorporated MOSTA data for E12.5 and E14.5 (new Supplementary Fig. 5a), which proved informative. At E12.5, MOSTA shows *Osr1* expression in connective tissues along the body axis, including the jaw region, lung primordium, meninges, mesothelium, muscle (likely corresponding to muscle connective tissue), and the urogenital ridge—consistent with previously published *Osr1* expression patterns from our group and others. Notably, a clear overlap with *Ebf2* was observed in the thoracic region corresponding to presumptive brown adipose tissue (BAT) progenitors. At E14.5, *Pparg* expression in MOSTA is detected in the dorsal thoracic region, overlapping with *Ebf2*, while *Osr1* expression is observed in the surrounding tissue of the presumptive early BAT anlage. This observation aligns closely with recent findings by Jun et al. (2023; Dev. Cell, doi:10.1016/j.devcel.2023.08.003), who identified a *Dpp4*⁺ adipogenic progenitor

population mapped to this same anatomical location in the MOSTA atlas, a finding we now also incorporated in the manuscript (altogether discussed on p.12, lines 305-312: “At E12.5, there was still widespread overlap of *Osr1* and *Ebf2* expression in anterior-dorsal regions mainly annotated as connective tissue (Supplementary Fig. 5a). At E14.5, *Pparg* expression was found at the presumptive site of BAT formation, here, *Ebf2* expression was found overlapping *Pparg*, while *Osr1* expression was preferentially seen in the circumference of *Pparg* expression (Supplementary Fig. 5a), again corroborating immunostaining data (Fig. 1b). Intriguingly, this expression domain overlaps with a pattern recently described by Jun et al⁴¹ for a signature of presumptive BAT progenitors characterized by expression of *Dpp4*. Of note, *Dpp4* expression is not detectable before E13.5⁴¹ and therefore not present in our E9.5 and E11.5 datasets (Supplementary Fig. 5b).”).

4. To build on the current findings, additional validation experiments showing how E9.5 *Osr1*⁺ cells contribute to adipogenic and non-adipogenic cells would be valuable. The experimental approach using only *Osr1*-EGFP positive cells is in my opinion not sufficient. The authors could consider using *Osr1*^{GCE}/+ mice crossed with Rosa26mTmG mice for single-cell analysis or FACS to demonstrate the multi-lineage potential of *Osr1*⁺ cells.

Thank you for this comment. We are not entirely certain how to interpret it, but we would like to clarify our approach. We used *Osr1*-eGFP expression from the *Osr1*^{GCE} line to identify *Osr1*⁺ cells *in vivo* and to perform comprehensive single-cell analysis of the *Osr1*⁺ population at each time point. In parallel, we employed the model mentioned by the reviewer (*Osr1*^{GCE/+} crossed with Rosa26^{mTmG} mice) for extensive, time-resolved lineage tracing of *Osr1*⁺ cells. These experiments clearly demonstrate the fibro-adipogenic potential of E11.5 *Osr1*⁺ cells and the broader multilineage potential of E9.5 *Osr1*-expressing cells *in vivo*. We think that performing single-cell RNA-seq on *Osr1*-lineage-traced cells would introduce an additional variable—tamoxifen-induced recombination efficiency. Given the low activity of the *Osr1*^{GCE} CreERT2 driver line, this approach would capture only a subset of the relevant population and thus might not yield comprehensive results.

We agree that differentiation assays using FACS-isolated *Osr1*-eGFP⁺ (or *Osr1*-lineage-labeled) cells could serve as a complementary line of evidence. In fact, we have made extensive attempts to culture *Osr1*⁺ cells from E9.5 and E11.5 embryos, but these efforts were unsuccessful. By contrast, *Osr1*⁺ cells from E13.5 embryos can be readily maintained in culture, as shown previously (Vallecillo-García et al., 2017; Nat. Commun. 8:1130; doi:10.1038/s41467-017-01120-3). We believe that the early *Osr1*⁺ populations are highly dependent on their *in vivo* microenvironment and cellular context. Even co-culture with feeder layers did not support their survival or expansion in our hands. Moreover, the number of *Osr1*⁺ cells available at E9.5 is extremely limited—approximately 1,500 cells per embryo—and only a fraction of these are expected to exhibit MAB-like properties according to our scSeq analysis, posing additional experimental challenges. We therefore consider our *in vivo* lineage tracing studies to provide stronger and more physiologically relevant evidence than would be achievable through *in vitro* differentiation assays. Nonetheless, we have added a statement acknowledging this limitation to the discussion section (p.21, lines 544-549: “We note that while our *in vivo* lineage-tracing data provide strong evidence for the multilineage differentiation potential of E9.5 *Osr1*⁺ cells, complementary *in vitro* validation would further strengthen these findings. Although we attempted to culture E9.5 *Osr1*⁺ cells according to protocols used for MABs^{23,46}, we were unable to successfully propagate them under the conditions tested. Further adaptations to perform cell culture differentiation experiments such as a three-dimensional context might be required⁵⁵.”).

5. It would be interesting to explore whether the distinct developmental origin of iBAT compared to cBAT and sBAT correlates with different functional roles or regulatory mechanisms. Would it be possible to analyze the existing data to predict certain features?

We agree that this represents an interesting and valuable perspective. However, to the best of our knowledge, no published studies or publicly available datasets currently compare different BAT depots in this context. Our impression is that the field generally regards the iBAT as the representative BAT depot in mice, and most studies have therefore focused on iBAT rather than other BAT depots.

In this regard, our findings provide additional evidence that there are notable developmental differences among BAT depots - an aspect that warrants further investigation and attention, which we had taken up in the manuscript (p.20 lines 503-505: "Moreover, this emphasizes that iBAT alone is not representative for BAT analysis. This is particularly relevant to humans, as the large iBAT depot present in human infants is not preserved in adults¹").

Reviewer #2 (Remarks to the Author):

This manuscript reports an alternative source of progenitor cells that give rise to brown adipocytes, the Osr1+ cells that are multipotent and appear at E9.5. This population is located in the dorsal aortic compartment and is proposed to be an early source of brown adipocytes during embryonic development. Interestingly, scRNA-seq data indicates that this population may correlate with MABs. The discovery of two distinct Osr1+ cell populations at E9.5 and E11.5 is quite interesting and potentially shows a new lineage of brown adipocytes that is time- and location-specific. I have a few comments that will hopefully improve the manuscript.

1. It would be nice to isolate Osr1+ cells at E9.5 and E11.5 to validate that these cells from E9.5 are multipotent, using differentiation protocols for brown adipocyte, white adipocyte, cartilage, and potentially muscle cells.

Thank you for the suggestion, which we agree would be a very good addition to the manuscript. We have in fact attempted this. While E13.5 Osr1+ cells can be easily isolated and differentiated *in vitro* (lineage restricted to adipocytes and fibroblasts, published in Vallecillo-Garcia et al. 2017 doi: 10.1038/s41467-017-01120-3), we have so far failed to culture Osr1+ cells isolated at either E11.5 or E9.5, including an attempt to cultivate the cells on feeder layers. Very likely, 3D cell context is required, however we think that trying to establish such a model would be beyond the scope of this study. We also note that from E9.5 embryos, only approx. 1500 cells can be isolated already posing limits to the experiment. Moreover, considering the single cell data, only a subset of our 1500 Osr1+ cells will represent the MAB-like behavior. We believe the *in vivo* lineage tracing studies we performed are superior to *in vitro* cell differentiation studies. We have now nevertheless added a statement reflecting this limitation to the discussion (p.21, lines 544-549: “We note that while our *in vivo* lineage-tracing data provide strong evidence for the multilineage differentiation potential of E9.5 Osr1+ cells, complementary *in vitro* validation would further strengthen these findings. Although we attempted to culture E9.5 Osr1+ cells according to protocols used for MABs^{23,46}, we were unable to successfully propagate them under the conditions tested. Further adaptations to perform cell culture differentiation experiments such as a three-dimensional context might be required⁵⁵.”).

2. Do Osr1+ cells at E9.5 reduce Osr1 expression at E11.5?

Thank you for this comment, this provided an interesting additional point towards the discussion of Osr1 cells' multipotency. We have analyzed the normalized expression level of Osr1 between datasets and now show an increase in Osr1 expression in Osr1-GFP cells between E9.5 and E11.5 (new Supplementary Fig. 6e). Such an upregulation of Osr1, which we and others have shown previously is able to suppress non-stromal cell fates, which is in line with our assumption of a higher level of multipotency of Osr1+ cells at E9.5 vs. E11.5. This is now elaborated in the manuscript, including novel data concerning the analysis of myogenic, tenogenic and chondrogenic signatures (new Supplementary Fig. 6c, d) on p.15 lines 364-377 (“To further investigate the lineage identities in our Osr1 dataset, we used embryonic mouse limb single cell sequencing data⁴³ and derived myogenic, tenogenic and chondrogenic module score signatures. While the myogenic signature exclusively mapped to E9.5 cl1 in line with the analysis above, tenogenic and chondrogenic signatures mapped predominantly to cl1 at E9.5 and shifted to cl0/2 at E11.5 (Supplementary Fig. 6c) in line with a hypothesized flow of cells in this direction. Expression of key marker genes that describe a tenogenic (Scx) as well as osteo-chondrogenic (Sox9 and Runx2) identity were detected in an overlapping pattern

(Supplementary Fig. 6d). Intriguingly, we have shown before that E13.5 Osr1+ cells still have expression of alternative lineage genes as Scx and Runx2, but that chondrogenic, tenogenic and also myogenic differentiation is actively repressed by Osr1 itself^{24,44}; indeed, Osr1 is a direct repressor of Sox9⁴⁵. Comparing Osr1 expression levels between the E9.5 and E11.5 datasets suggested an upregulation of Osr1 over time as cells progress to an overall stromal fate (Supplementary Fig. 6e). This would agree with the assumption that a lower level of Osr1 expression in Osr1+ cells at E9.5 may be instrumental to maintain a multipotent fate, which becomes more and more restricted as Osr1 expression increases.”).

3. Perilipin staining could be added as a more specific marker for adipocytes, at least in the major images showing BAT, as FABP4 is expressed in many other cell types. Also, UCP1 could be used to label brown adipocytes.

We thank the reviewer for this important suggestion. We have performed Perilipin and UCP1 immunostaining on E18.8 Osr1^{GCE}/R26^{mTmG} embryos Tamoxifen-induced at E9.5 or E11.5. **New Supplementary Figures 1e and 3d** show representative images from the sBAT depot clearly demonstrating BAT identity of Osr1-lineage cells from both time points.

4. Please include more discussion about the differences observed in different BAT depots. Do they potentially have different functions in thermogenesis/metabolism?

We appreciate that the reviewer also highlights this interesting aspect. As mentioned in our response to Reviewer #1, to the best of our knowledge there are currently no published studies or datasets that systematically compare different BAT depots in mice. In this context, our results further underscore that distinct BAT depots may differ developmentally, suggesting that depot-specific origins and characteristics, which we had taken up in the manuscript (p.20 lines 503-505: “Moreover, this emphasizes that iBAT alone is not representative for BAT analysis. This is particularly relevant to humans, as the large iBAT depot present in human infants is not preserved in adults¹.”).

Reviewer #3 (Remarks to the Author):

The manuscript provides a compelling and innovative perspective on BAT developmental biology by identifying the dorsal aortic compartment as a novel source of BAT progenitors. The use of advanced tools, including single-cell RNA sequencing, lineage tracing, and immunohistochemistry, adds rigor and credibility to the findings. The hypothesis that E9.5 *Osr1*⁺ progenitors are distinct from dermomyotomal progenitors is clearly articulated and well-supported. However, additional functional validations would strengthen the manuscript and further support the conclusions. The study convincingly describes the migratory behavior and multipotency of E9.5 *Osr1*⁺ progenitors, yet the functional evidence for their differentiation potential remains limited as well their implication in controlling adipogenesis. To further enhance the manuscript, I recommend the following:

1) Validate Multipotency of E9.5 *Osr1*⁺ Progenitors: Perform *in vitro* differentiation assays to test the capacity of these progenitors to differentiate into adipogenic, myogenic, and other mesodermal lineages. Alternatively, conduct transplantation experiments to evaluate their contribution to BAT and other lineages *in vivo*.

Thank you for the valuable suggestion, which we agree would represent a meaningful addition to the manuscript. We have indeed attempted *in vitro* differentiation assays. While *Osr1*⁺ cells isolated at E13.5 can be readily cultured and differentiated *in vitro* (lineage-restricted to adipocytes and fibroblasts, as reported in Vallecillo-Garcia et al., 2017; doi: 10.1038/s41467-017-01120-3), our efforts to culture *Osr1*⁺ cells obtained at earlier developmental stages (E11.5 or E9.5), including attempts using feeder layers, have been unsuccessful. Likely, the limited number of *Osr1*⁺ cells obtainable from E9.5 embryos (~1,500 cells, only a small fraction of which will represent MAB-like cells according to our scSeq data) imposes experimental constraints. While we consider that the *in vivo* lineage tracing analyses we performed provide a more robust and physiologically relevant assessment than *in vitro* differentiation approaches, we have now added a statement acknowledging this limitation to the discussion (p.21, lines 544-549: “We note that while our *in vivo* lineage-tracing data provide strong evidence for the multilineage differentiation potential of E9.5 *Osr1*⁺ cells, complementary *in vitro* validation would further strengthen these findings. Although we attempted to culture E9.5 *Osr1*⁺ cells according to protocols used for MABs^{23,46}, we were unable to successfully propagate them under the conditions tested. Further adaptations to perform cell culture differentiation experiments such as a three-dimensional context might be required⁵⁵.”).

2) Assess Thermogenic Potential of BAT derived from *Osr1*⁺ progenitors: Evaluate the thermogenic activity of BAT derived from E9.5 *Osr1*⁺ progenitors using molecular markers like UCP1 or through metabolic assays could address this point.

We have now performed UCP1 and Perilipin immunostaining on tissue sections of *Osr1*^{GCE}/*R26*^{mTmG} E18.5 embryos, which show that both, E9.5 (new Supplementary Fig. 3d) and E11.5 (new Supplementary Fig. 1e) *Osr1*-lineage cells contribute to bona-fide BAT cells.

3) Clarify the Role of *Osr1* in Adipogenesis: Use *Osr1* knockout or knockdown models to investigate its specific role in regulating adipogenesis. Examine whether *Osr1*⁺ progenitors include subpopulations that overlap with CD142⁺ adipogenesis-regulatory cells (*Areg*) previously identified in adult adipose

depots or mesoangioblasts (Schwalie et al., Nature 2018, 559(7712):103-108; Camps et al., Cell Rep. 2020, 31(5):107597)

Thank you for these thoughtful suggestions, which we agree raise important points. Concerning the use of *Osr1* KO models, we consider that the current manuscript represents a complete and coherent study with a focused message, specifically addressing the lineage relationships underlying brown adipose tissue development. Incorporating phenotypic knockout data would introduce an additional level of complexity that cannot be explored in sufficient depth within the current framework without unduly inflating its content. We have, in fact, already collected data on the adipose tissue phenotype of *Osr1* knockout fetuses, which could be included if deemed essential. Nonetheless, we are presently conducting a detailed analysis of the molecular functions of *Osr1* in preadipogenic cells. We therefore believe that these mechanistic findings are best presented in a separate manuscript that specifically focuses on elucidating the role of *Osr1* in preadipogenic differentiation. In the current study, we aim to maintain our focus on employing *Osr1* as a marker to define the developmental origins of adipogenic progenitors.

Concerning CD142+ AREGS, this is indeed a very interesting point, as AREGs show prominent expression of *Osr1* as shown in the study by Schwalie et al. Supplementary Fig. 4k. We have analyzed *F3*/CD142 expression in our dataset and also compared the expression of *Osr1* and *F3*/CD142 expression in the Angueira (doi: 10.1038/s42255-021-00380-0) dataset (new Supplementary Fig. 8) and included a paragraph of discussion to this end (p.22 lines 568-577: “Moreover, the progenitor populations in adult subcutaneous WAT have been further stratified, leading to the discovery of a specific population of adipogenic-regulatory cells (AREGS) characterized by expression of *F3* encoding CD142⁵⁶. A similar population has been identified in intramuscular adipose tissue⁵⁷. AREGS suppress adipogenesis in a paracrine manner and intriguingly show high expression of *Osr1*^{56,57}. *F3* expression was not detectable in our E9.5 dataset, at E11.5 *F3* was detected in cI0/2 in a dispersed pattern not matching the pre-BAT module (Supplementary Fig. 8a). It remains unknown, however, if *F3*/CD142+ cells function as AREGS in the embryo. Comparing *Osr1* and *F3* expression in the aortic BAT datasets²⁷ confirmed partial overlap, yet *Osr1* showed a broader expression suggesting that *Osr1* is not an exclusive AREG marker but plays a broader role (Supplementary Fig. 8b)”.

These additional experiments will substantiate the authors' conclusions and further elucidate the role of *Osr1*+ progenitors in BAT development and adipogenesis.